# Comparative Phylogenomic Analysis Reveals Evolutionary Genomic Changes and Novel Toxin Families in Endophytic *Liberibacter* Pathogens

Yongjun Tan,[a] Cindy Wang,[a] Theresa Schneider,[a] Huan Li,[a] Robson Francisco de Souza,[b] Xueming Tang,[c] Kylie D. Swisher Grimm,[d] Tzung-Fu Hsieh,[e,f] ⓘ Xu Wang,[g,h,i] Xu Li,[e,f] ⓘ Dapeng Zhang[a,j]

[a]Department of Biology, College of Arts & Sciences, Saint Louis University, St. Louis, Missouri, USA
[b]Departamento de Microbiologia, Instituto de Ciências Biomédicas, Universidade de São Paulo, São Paulo, Brazil
[c]School of Agriculture and Biology, Shanghai Jiao Tong University, Shanghai, China
[d]United States Department of Agriculture—Agricultural Research Service, Temperate Tree Fruit and Vegetable Research Unit, Prosser, Washington, USA
[e]Department of Plant and Microbial Biology, North Carolina State University, Raleigh, North Carolina, USA
[f]Plants for Human Health Institute, North Carolina State University, Kannapolis, North Carolina, USA
[g]Department of Pathobiology, College of Veterinary Medicine, Auburn University, Auburn, Alabama, USA
[h]Alabama Agricultural Experiment Station, Auburn University, Auburn, Alabama, USA
[i]HudsonAlpha Institute for Biotechnology, Huntsville, Alabama, USA
[j]Bioinformatics and Computational Biology Program, College of Arts & Sciences, Saint Louis University, St. Louis, Missouri, USA

**ABSTRACT** *Liberibacter* pathogens are the causative agents of several severe crop diseases worldwide, including citrus Huanglongbing and potato zebra chip. These bacteria are endophytic and nonculturable, which makes experimental approaches challenging and highlights the need for bioinformatic analysis in advancing our understanding about *Liberibacter* pathogenesis. Here, we performed an in-depth comparative phylogenomic analysis of the *Liberibacter* pathogens and their free-living, nonpathogenic, ancestral species, aiming to identify major genomic changes and determinants associated with their evolutionary transitions in living habitats and pathogenicity. Using gene neighborhood analysis and phylogenetic classification, we systematically uncovered, annotated, and classified all prophage loci into four types, including one previously unrecognized group. We showed that these prophages originated through independent gene transfers at different evolutionary stages of *Liberibacter* and only the SC-type prophage was associated with the emergence of the pathogens. Using ortholog clustering, we vigorously identified two additional sets of genomic genes, which were either lost or gained in the ancestor of the pathogens. Consistent with the habitat change, the lost genes were enriched for biosynthesis of cellular building blocks. Importantly, among the gained genes, we uncovered several previously unrecognized toxins, including new toxins homologous to the EspG/VirA effectors, a YdjM phospholipase toxin, and a secreted endonuclease/exonuclease/phosphatase (EEP) protein. Our results substantially extend the knowledge of the evolutionary events and potential determinants leading to the emergence of endophytic, pathogenic *Liberibacter* species, which will facilitate the design of functional experiments and the development of new methods for detection and blockage of these pathogens.

**IMPORTANCE** *Liberibacter* pathogens are associated with several severe crop diseases, including citrus Huanglongbing, the most destructive disease to the citrus industry. Currently, no effective cure or treatments are available, and no resistant citrus variety has been found. The fact that these obligate endophytic pathogens are not culturable has made it extremely challenging to experimentally uncover the genes/proteins important to *Liberibacter* pathogenesis. Further, earlier bioinformatics studies failed to identify key genomic determinants, such as toxins and effector proteins, that underlie the pathogenicity of the bacteria. In this study, an in-depth comparative genomic analysis of *Liberibacter*

Address correspondence to Dapeng Zhang, dapeng.zhang@slu.edu.

🐦 @DapengZhang5

pathogens along with their ancestral nonpathogenic species identified the prophage loci and several novel toxins that are evolutionarily associated with the emergence of the pathogens. These results shed new light on the disease mechanism of *Liberibacter* pathogens and will facilitate the development of new detection and blockage methods targeting the toxins.

**KEYWORDS** *Liberibacter* pathogens, Huanglongbing, zebra chip, toxins, prophages, pathogenesis, evolution, comparative genomics

"Candidatus Liberibacter," a genus of Gram-negative bacteria in the order of *Rhizobiales*, has recently received increasing attention because several of its species are closely associated with severe diseases in multiple crop plants, such as citrus, potato, tomato, and carrot. These include "*Candidatus* Liberibacter asiaticus" found in Asia and North America, "*Candidatus* Liberibacter africanus" found in Africa (1), and "*Candidatus* Liberibacter americanus" found in Brazil (2) that cause the citrus Huanglongbing (HLB) disease and "*Candidatus* Liberibacter solanacearum" that causes similar diseases in tomato (psyllid yellows disease), potato (zebra chip [ZC] disease), and carrot around the world (3, 4).

HLB disease, also known as citrus greening, is the most destructive worldwide disease of citrus (5). It is characterized by yellowing of citrus leaves, premature defoliation, decay of feeder rootlets and lateral roots, production of small, bitter fruit, and eventually death of the citrus tree (5). In all, HLB has led to a substantial loss of citrus production and severe damage to the economy and job market (6). Unfortunately, there is currently no effective cure or treatment available, and no resistant citrus variety has been found so far. As a result, infected citrus trees are often abandoned, creating a hot spot for the pathogens and their insect vectors and perpetuating the problem in citrus-growing regions around the world.

Likewise, ZC is a new disease of potato that was first identified in Mexico in 1994 (7) and has quickly spread to many countries in North and South America and New Zealand in recent years (8). ZC negatively affects growth, yield, and quality of tubers and thus significantly affects international trade and the economy (9). Similar to HLB, there is currently no effective treatment for "*Ca*. Liberibacter solanacearum" infection. Instead, growers rely on monitoring vector populations and treating with different insecticides. There is also no resistant potato variety currently in production. The pathogen is equally as destructive to Apiaceae crops like carrot and celery in Europe, the Mediterranean region, and northern Africa, causing yield and quality defects that render crops unmarketable (8, 10).

Given the severity and rapid spreading of these diseases in commercial crops, extensive research efforts have been made to better understand the basic biology of these pathogens and the pathogenesis mechanisms of the diseases. However, the progress has been slow due to several main obstacles. *Liberibacter* pathogens are endophytic bacteria transmitted naturally by several psyllid vectors, such as Asian citrus psyllid (*Diaphorina citri*) (11), African citrus psyllid (*Trioza erytreae*) (12), potato psyllid (*Bactericera cockerelli*) (9), and carrot psyllids (*Trioza apicalis*) (13, 14). This, along with them being unculturable, makes controlled inoculation for studying host-pathogen interaction extremely difficult (15). Moreover, secreted toxins or effectors, either manipulating host/vector immunity and physiology or damaging the host cells (16), represent the most important pathogenicity determinants of bacterial pathogens. Yet, identities of the toxins or effectors in *Liberibacter* pathogens have remained elusive for years. Previous research has relied mostly on bioinformatics prediction of secreted proteins carrying T2SS-specific signal peptides followed by functional examination (17–19). However, such strategy suffers from predicting too many candidates, making functional validation inefficient or impractical. Further, the assumption that HLB-associated toxins/effectors contain a signal peptide is premature and might not necessarily be valid. Thus far, the potential toxins/effectors identified by earlier studies are limited to a small number of HLB-associated pathogens. For instance, CLIBASIA_03875 (m3875) (20) is present in only one "*Ca*. Liberibacter asiaticus" strain; other reported toxins, such as Sec-delivered effector 1 (SDE1; CLIBASIA_05315) (21, 22), Las5315mp (23), SDE15 (CLIBASIA_04025) (24), and CLIBASIA_04405 (m4405) (25), are present only in "*Ca*. Liberibacter asiaticus" stains but not in other HLB-associated pathogens ("*Ca*. Liberibacter americanus" and

"*Ca.* Liberibacter africanus"). This suggests that other types of unidentified toxins or effectors might be responsible for the primary HLB pathology. Consequently, without accurate and comprehensive information about the HLB-associated toxins/effectors, targeted strategies for detection, prevention, and blockage will not be possible.

In this study, we aimed to tackle these problems by utilizing available genome information and dedicated bioinformatics strategies. Our analysis is based on the observed phenotypic difference between the most ancestral *Liberibacter* species, *Liberibacter crescens*, which displays a free-living habitat and is apparently nonpathogenic, and the descendants that are both endophytic and pathogenic. We hypothesized that the functional difference is determined by the genomic changes, including both gene-loss and gene-gain events, that occurred at the common ancestor of the pathogens. We designed multiple comparative genomics strategies to systematically mine the major genomic differences, including unique prophage loci and other genes that are evolutionarily associated with the emergence of the pathogens. More importantly, we identified several potential toxin proteins, including novel toxins homologous to the EspG/VirA effectors, a YdjM phospholipase toxin, and a secreted endonuclease/exonuclease/phosphatase (EEP) family protein. The new information gained from this research provides important insight into the evolution and pathogenesis of *Liberibacter* pathogens and will facilitate development of novel detection and blockage methods targeting the toxins.

## RESULTS

**Phylogenetic relationships of HLB-associated pathogens suggest an evolutionary transition from nonpathogenic ancestor to pathogenic descendants.** We first sought to understand the evolutionary relationship of the HLB-associated pathogens. We collected all *Liberibacter* species whose genome information is available in the NCBI GenBank database (Fig. 1A and Table S1). This includes eight genomes of HLB-associated pathogens (six strains of "*Ca.* Liberibacter asiaticus" [26–31], one genome of "*Ca.* Liberibacter africanus" [32], and one of "*Ca.* Liberibacter americanus" [2]), one genome of "*Ca.* Liberibacter solanacearum" (33), one genome of "*Candidatus.* Liberibacter europaeus" (which is a potential pathogen of *Cytisus scoparius* and vectored by *Arytainilla spartiophila*) (34, 35), and two genomes of *Liberibacter crescens* (which was isolated from papaya and represents the most basal lineage within the *Liberibacter* genus) (36, 37). Maximum likelihood phylogenetic analyses of three conserved genes, 16S rRNA, 23S rRNA, and DNA polymerase I, support the same tree topology (Fig. 1B to D). Importantly, this analysis revealed that the species associated with HLB are not monophyletic. Both "*Ca.* Liberibacter africanus" and "*Ca.* Liberibacter asiaticus" share a common ancestor, whereas "*Ca.* Liberibacter americanus" shows a close relationship with "*Ca.* Liberibacter europaeus." Between these two groups is "*Ca.* Liberibacter solanacearum," a pathogen of potatoes, tomatoes, and carrots. Further, all five species of pathogens share a common ancestor with *L. crescens* as a sister group. *L. crescens* bacteria are free living and do not seem to be pathogenic, despite the fact that they were isolated from papaya (37).

The phyletic pattern between the ancestral species and the descendants revealed a clear demarcation between the free-living nonpathogenic *Liberibacter* bacteria and the endophytic pathogenic species, suggesting that both the intracellular habitat and pathogenicity are derived traits of *Liberibacter* during evolution. We hypothesized that the development of these new traits was attributed to genomic changes, including both gene-loss and gene-gain events, that occurred in the ancestor of the *Liberibacter* pathogens. Based on this hypothesis, we designed several phylogenomic comparison strategies to specifically identify genes or genomic regions that display unique phyletic patterns in either pathogenic descendants or ancestral *L. crescens* species to understand the evolution and pathogenicity mechanisms of *Liberibacter*.

**Whole-genome comparisons of *Liberibacter* bacteria.** The first computational strategy involved comparing whole *Liberibacter* genomes in order to develop a general idea of changes and dynamics of large genomic regions during evolution. Pairwise TBLASTX comparisons were used to identify gene correspondence between the genomes (indicated by direct lines in Fig. 2). We found that between five pathogenic species, the majority of their genome regions preserve similar gene composition and organization, with several

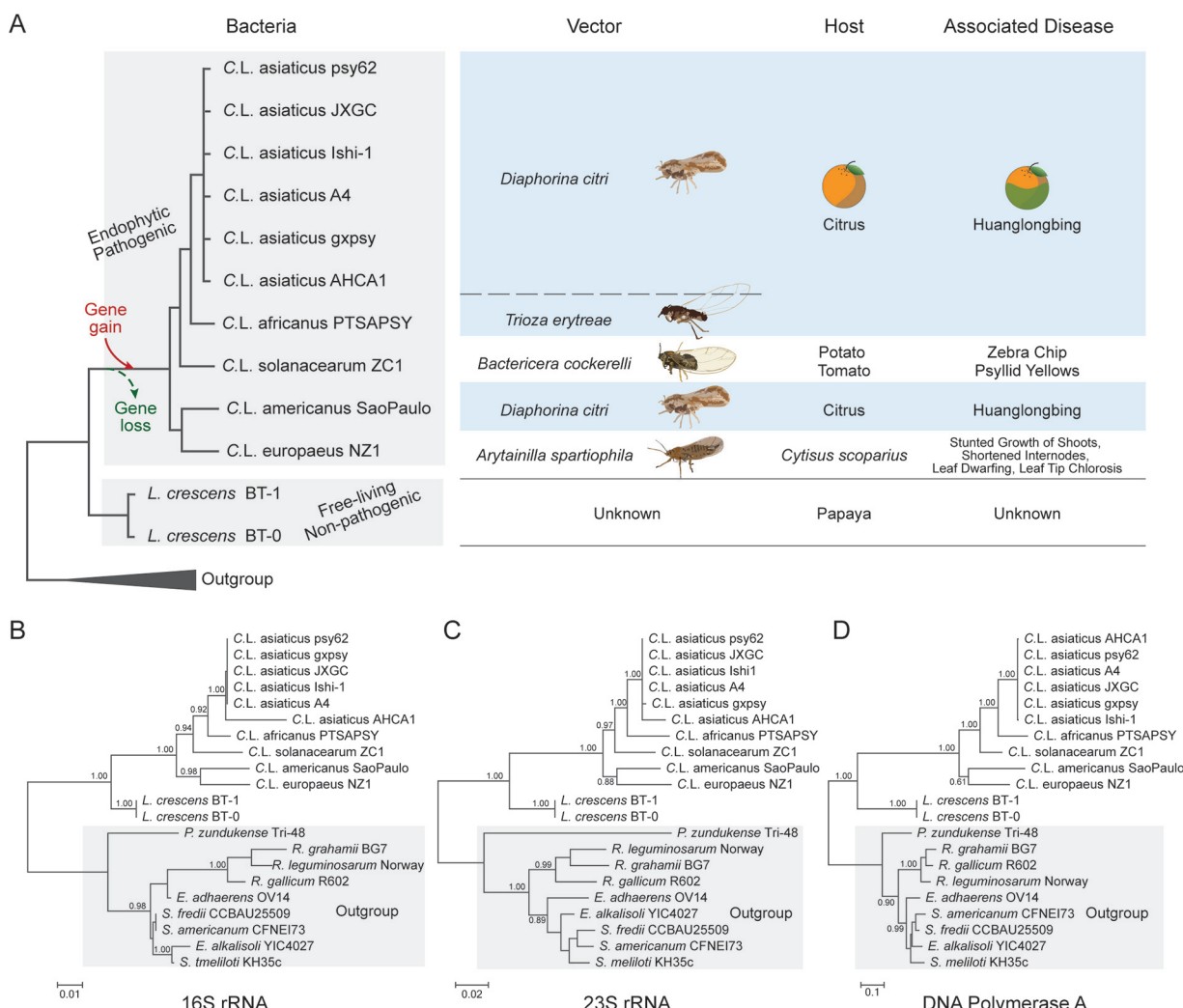

**FIG 1** Phylogenetic relationship of HLB-associated bacteria and their relative species. (A) Detailed information of *Liberibacter* species whose genomes were investigated in this study, including their evolutionary relationship, transmitted vectors, hosts, and associated diseases. The evolutionary relationship was derived from phylogenetic trees of 16S rRNA (B), 23S rRNA (C), and DNA polymerase I (D). (B to D) The phylogenetic trees were inferred using the maximum likelihood (ML) method, where the supporting values from 100 bootstrap are shown for major branches only. The outgroup clades have a gray background.

large-scale genome inversions (as shown in crossed blue lines in Fig. 2). When we compared genomes of the nonpathogenic *L. crescens* and pathogenic "*Ca.* Liberibacter europaeus," their genome organization was more diverse than that between genomes of pathogens. This suggests that extensive genomic changes did occur between the nonpathogenic ancestor and pathogenic ancestor, consistent with their transitions in living habitat and pathogenicity.

**Genomic organization dissection and classification of *Liberibacter* prophage loci.** Prophage loci are typically among the most variable regions on bacterial genomes and play a critical role in pathogenesis (38) by carrying toxins or virulence factors, such as cholera toxins (39) and the toxins used by pathogenic *Escherichia coli* (40). Indeed, Figure 2 shows a striking difference in length of prophage loci on these genomes. Although several prophage loci in *Liberibacter* bacteria were identified previously (29, 36, 41, 42), a clear understanding of their genomic organizations, functional compositions, evolutionary relationships, and origins remains missing. Thus, we conducted an in-depth gene neighborhood analysis to systematically extract the potential prophage loci, identify their boundaries, and annotate their components. As a result, we retrieved 36 prophage loci from the 12 available *Liberibacter* genomes (Fig. 3, Table S2, and Data

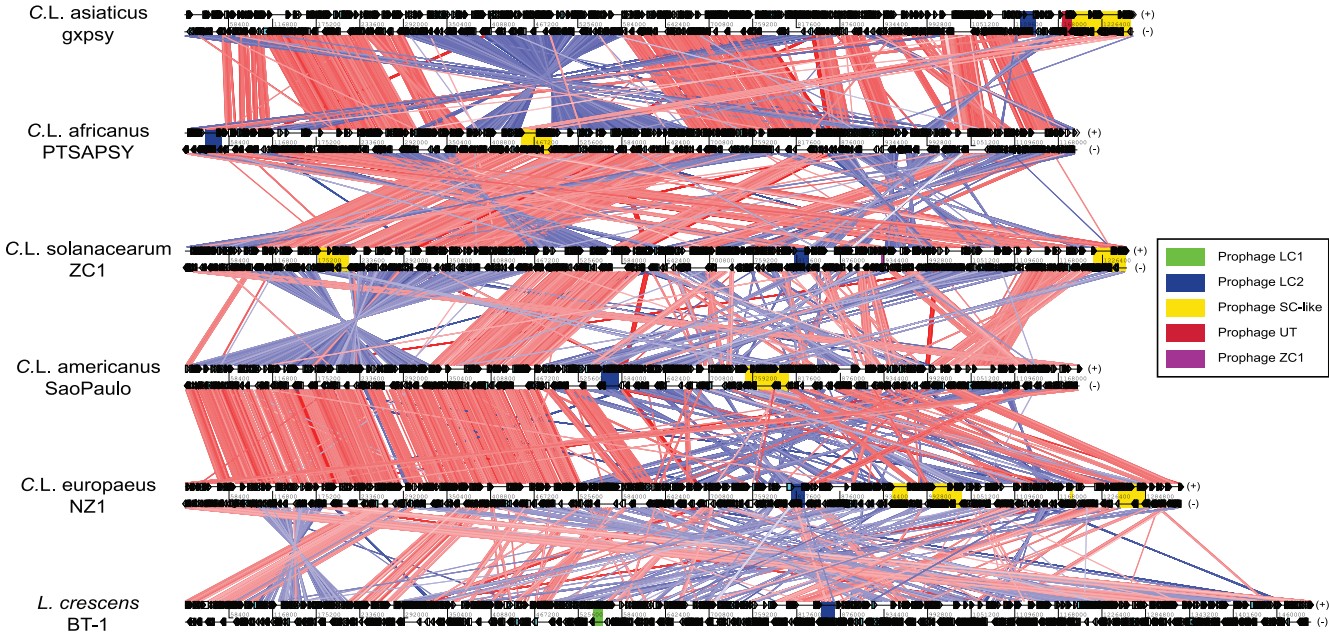

**FIG 2** Whole-genome comparisons of *Liberibacter* species. The genomes are labeled by their scientific names followed by strain information. Cyan and yellow boxes are the protein coding frames and RNA regions on either forward (+, top) or reverse (−, bottom) strands, respectively. Homologous genes between genomes are linked by lines, where the red and blue lines represent the forward and reverse (complementary) matches, respectively. The intensity of the color bands is proportional to the percent identity of the match, where higher intensity indicates higher sequence identity. The phage loci are shown in colored boxes (legend on the right).

Set S1). We annotated their gene components by clustering the protein products and dissecting their domains. We found that these prophage loci displayed a highly variable gene composition. However, based on the shared gene components, we were able to classify them into four major types, including LC1, LC2, SC, named according to the representative prophage locus identified in previous studies (36, 41), and UT, a unique type of prophage (Fig. 3). Specifically, the LC1 type is only found in nonpathogenic *L. crescens* whereas LC2 is present in all *Liberibacter* species. They are similar in terms of gene composition; however, LC2 contains several unique genes, such as LC_TM, Peptidase_S74_2, LC_3, HTH_XRE, and DUF1376. The SC type is found in all *Liberibacter* pathogens and typically has a large genome size with big variations among loci which are caused mainly by independent genome deletion and insertions. It is noteworthy that several genomes contain multiple copies of the SC-type phages. These include two copies in "*Ca.* Liberibacter asiaticus" gxpsy genome, two copies in "*Ca.* Liberibacter solanacearum" ZC1 genome, and three copies in "*Ca.* Liberibacter europaeus" NZ1 genome. In contrast, the UT type represents a novel group of prophages that we recovered in all "*Ca.* Liberibacter asiaticus" strains. The size of the UT type is much smaller, and its gene composition is close to that of the SC type. There is also a small prophage locus (ZC1) on the "*Ca.* Liberibacter solanacearum" strain ZC1 genome which might be the ancient prophage remnant after excision from the host genome.

**Phylogenetic analysis reveals independent gene transfers of phages to *Liberibacter*.** We next attempted to trace the evolutionary histories of these prophages to examine their correlations with the development of pathogenicity. As terminase, the key phage component involved in the phage DNA packing process (43), is the only gene component conserved across all identified prophage loci (Fig. 3), it was used as a marker to infer the evolutionary origins of these prophages. We conducted several BLASTP searches using different terminases from *Liberibacter* species to collect homologs. Using both maximum likelihood and Bayesian inference analyses (Fig. 4A), we found that the terminases of *Liberibacter* prophages are not monophyletic; instead, they are nested in three separate clades. This indicates that the LC1, LC2, and SC (together with UT) prophages have different evolutionary origins and that they have been transferred independently to *Liberibacter* at different evolutionary time points (Fig. 4A). In addition to their independent origins, the tree also suggests that multiple copies

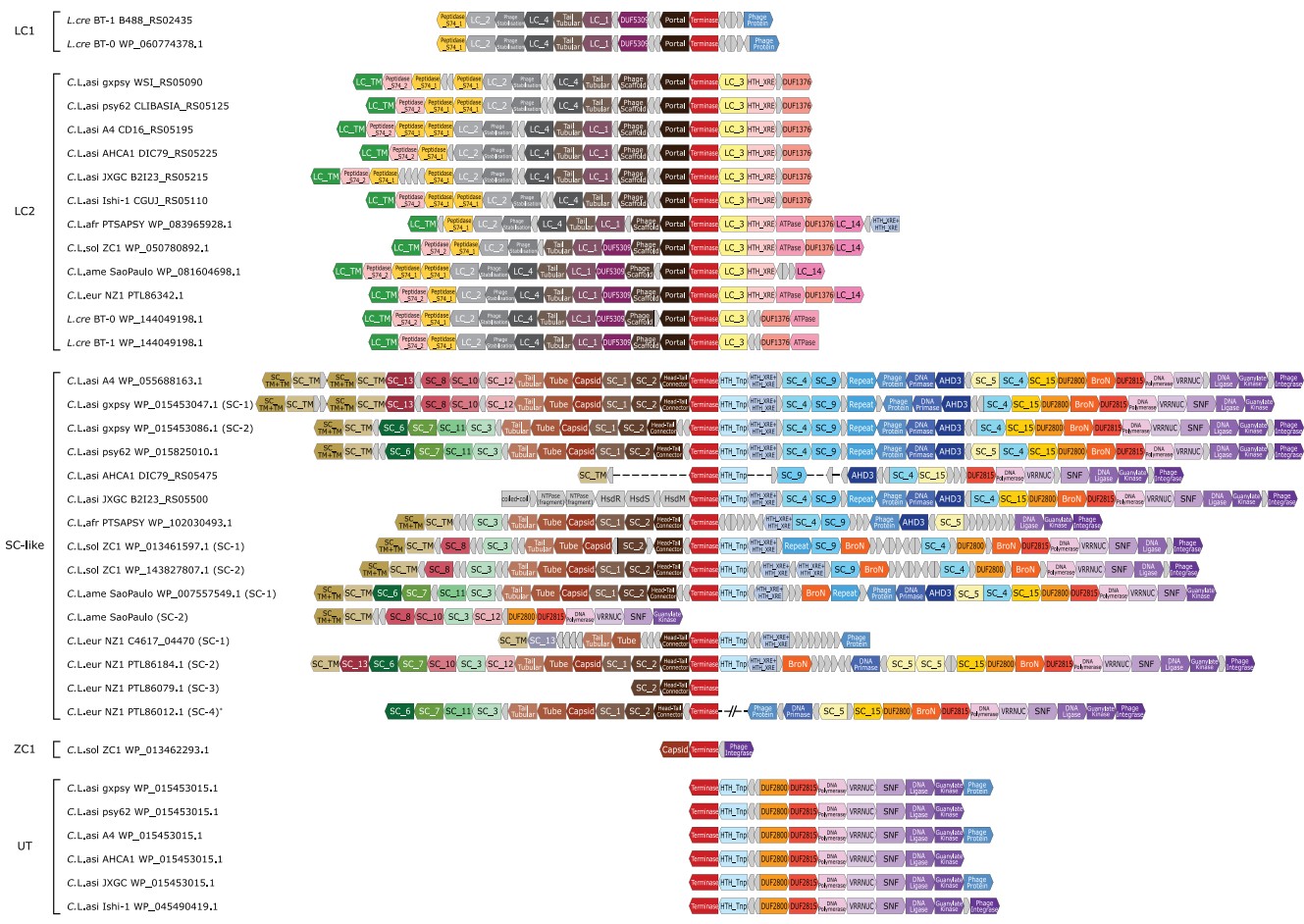

**FIG 3** Genomic structures of the prophages identified in the *Liberibacter* species. The coding frames in prophage loci are presented in blocks. The blocks, labeled with gene annotations and highlighted in different colors, are the coding products shared by at least four prophage loci, while the small gray blocks represent nonconserved coding products or pseudogenes. The genome structures were aligned based on the shared terminases of these prophage loci. On the left side, the prophage loci are indicated by their species names, strains, and terminase accession numbers or locus tags. The prophage loci were classified into four types (LC1, LC2, SC-like, UT) and a distinct prophage fragment found in a "*Ca.* Liberibacter solanacearum" ZC1 strain. One "*Ca.* Liberibacter europaeus" SC-like prophage structure was derived from two genome contigs, PSQJ01000015.1 and PSQJ01000003.1, which is indicated by an asterisk (*).

of the SC phage in several *Liberibacter* pathogens were likely generated from genome-specific duplications, such as those in "*Ca.* Liberibacter solanacearum," "*Ca.* Liberibacter europaeus," and "*Ca.* Liberibacter asiaticus" (Fig. 4A). However, the SC prophages appear to have undergone further diversification and recombination, given the following facts: (i) their terminases do not follow a typical pattern of vertical evolution, unlike the LC2 terminases (Fig. 4A), and (ii) in the genome comparisons, the SC phages from different "*Ca.* Liberibacter asiaticus" strains display a striking divergence in certain regions, in contrast to other genomic regions (Fig. 4B).

Based on the results from gene neighborhood classification and phylogenetic analysis, we propose that LC2, LC1, and SC prophages were introduced at the base of *Liberibacter*, the base of nonpathogenic *L. crescens*, and the common ancestor of the *Liberibacter* pathogens, respectively (Fig. 4C). As for the UT phage, it likely originated from a duplication event of the ancestral SC phage at the base of the "*Ca.* Liberibacter asiaticus" species (Fig. 4B and C). According to the phylogenetic histories of these prophages, we can infer that the origin of the SC type of prophages was associated with the emergence of pathogenicity of the *Liberibacter* decedents.

**Ortholog clustering analysis reveals additional unique genes which were either lost or gained in the ancestor of pathogens.** We next sought to identify other genomic (nonprophage) genes which might be associated with the transition from the nonpathogenic ancestor to pathogenic descendants. These genes should be featured by having undergone either gene-loss or gene-gain events in the common ancestor of all

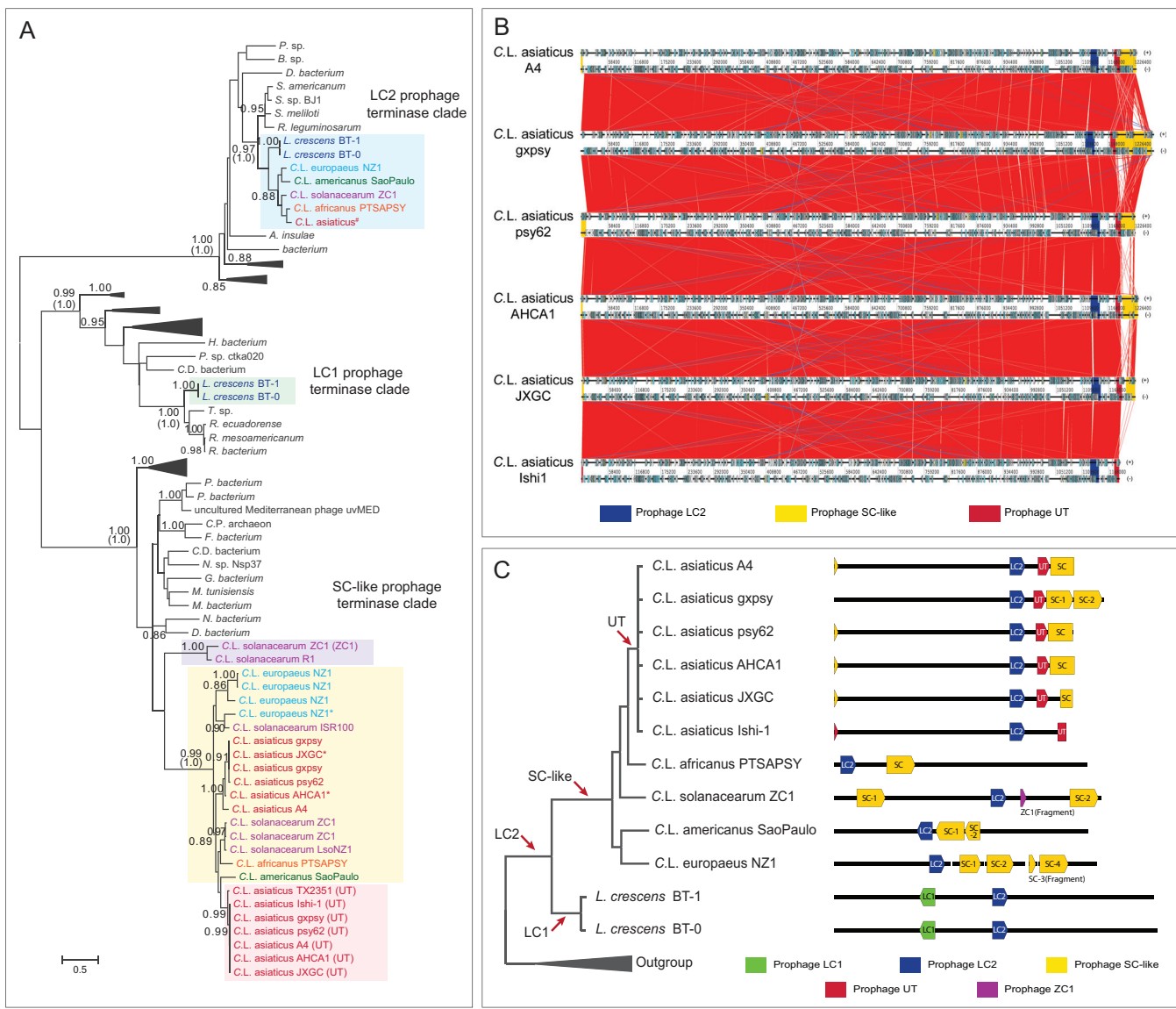

**FIG 4** Independent origins of *Liberibacter* prophages. (A) Phylogenetic relationship of prophage terminases. The phylogeny was constructed by using maximum likelihood (ML) and Bayesian inference (BI) methods. The ML tree with the highest log-likelihood is presented with the major branches supported by both ML bootstrap values (top) and the BI posterior values (bottom, bracketed). The terminases from the previously defined *Liberibacter* prophage types are highlighted in different background colors: LC2 prophage in light blue, LC1 prophage in light green, ZC1 prophage in light purple, UT prophage in light red, and SC prophage in yellow. Sequence annotations were colored differently according to their species. The terminase sequences from "*Ca.* Liberibacter asiaticus" strains are identical; therefore, one sequence is used to represent them and is indicated by a hashtag (#). Several protein sequences were translated by the ORFfinder program, which are indicated by an asterisk symbol (*). (B) Whole-genome comparison of different strains of "*Ca.* Liberibacter asiaticus." Homologous genes between genomes are linked by lines. Colored boxes are the prophage loci, and the high variation of SC-type prophage loci is shown at the 3′ end of the genomes. (C) The inferred independent origins of different *Liberibacter* phage types. The evolutionary relationship of *Liberibacter* species is shown on the left, and their compositions of the phage loci are illustrated on the right. The color key is shown on the bottom. The red arrows indicate the potential gene transfer events of different prophage types.

pathogens. Therefore, we were specifically interested in identifying two groups of genes which display unique phyletic patterns: first, the genes that underwent gene loss are present in the nonpathogenic *L. crescens* species but not in pathogenic descendants, and second, the genes that underwent gene gain are present in all pathogens but not in nonpathogenic *L. crescens* species (Fig. 5A).

To identify these genes, we utilized three steps of analysis. First, we carried out an ortholog analysis using the OrthoFinder program (44) to cluster the proteins (excluding the prophage components) of all collected *Liberibacter* genomes into different orthologous groups (orthogroups). Based on their phyletic profiles, we identified the orthogroups that

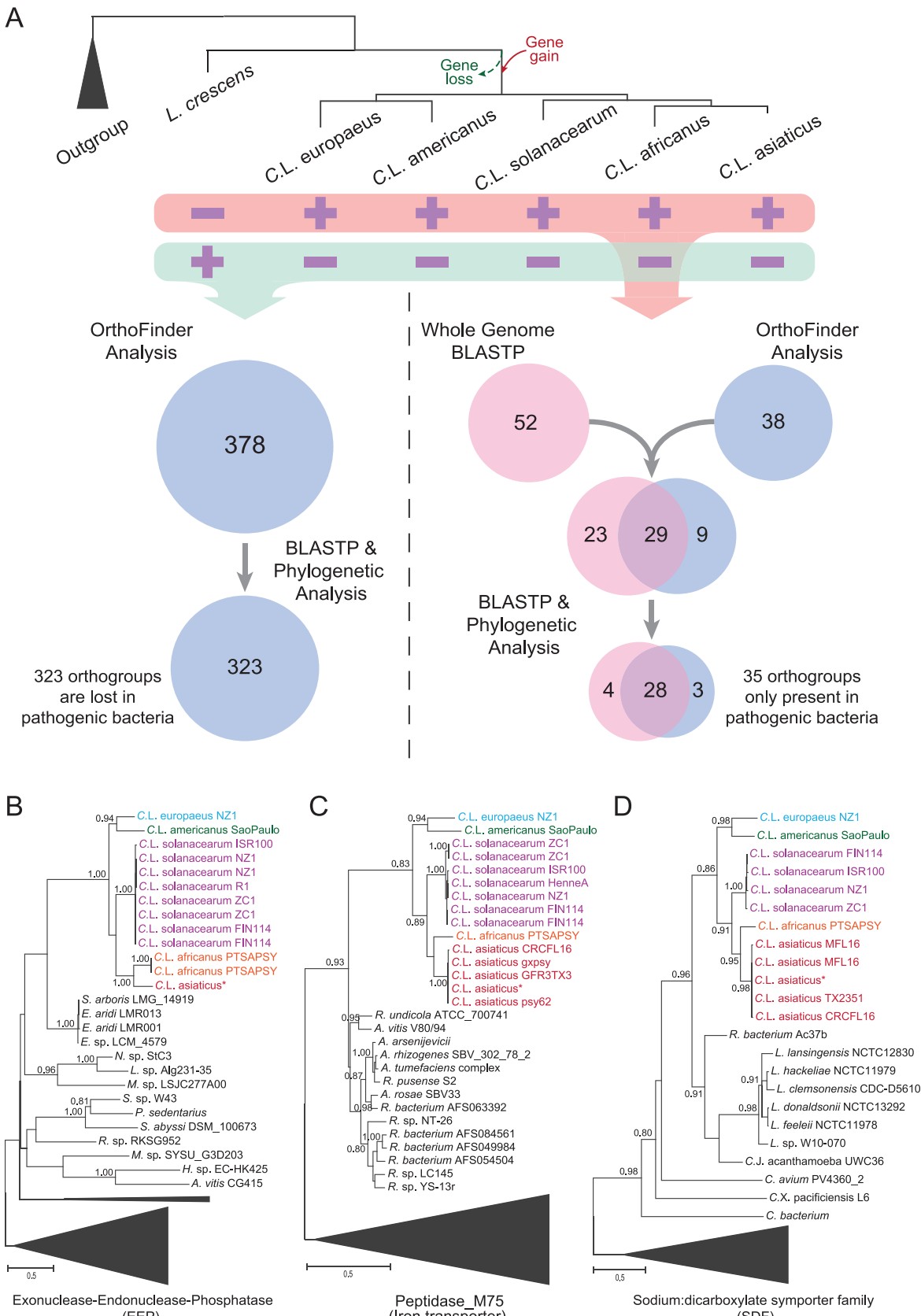

**FIG 5** Ortholog clustering analysis identified the genomic genes that were gained and lost during the transition from nonpathogenic ancestor to pathogenic descendants. (A) A conceptual figure illustrating the strategies to identify the *Liberibacter* genes which were

are present only in pathogens (but not in the nonpathogenic ancestor) or only in the nonpathogenic ancestor (but not in pathogens). We also utilized whole-genome BLASTP searches to specifically identify the cases of gene gain or gene loss by comparing the *Liberibacter* proteins with other sequences in the NCBI database, according to the pairwise sequence similarity scores. Finally, to validate and confirm the above results, we conducted extensive phylogenetic analyses on all identified orthogroups. From these analyses, we identified 323 orthogroups (about 335 genes in *L. crescens* BT-1) that were lost in all pathogenic descendants and 35 orthogroups (about 35 to 37 genes in each pathogen) that were gained in the ancestor of all pathogens (Fig. 5A and Table S3 and S4). Figure 5B to D shows the evolutionary histories of the three orthogroups. The tree topology supports a common ancestry of the homologous genes at the base of the *Liberibacter* pathogens, and their origins were likely from bacteria other than the nonpathogenic *Liberibacter* ancestor.

**The gene-loss and gene-gain events might contribute to the establishment of endophytic habitats of pathogens.** To understand the functional significance of these two types of orthogroups, we performed Gene Ontology and KEGG pathway analysis. This revealed that many of the genes that were lost in the endophytic pathogens are involved in synthesis and metabolism of several major types of cellular components, including amino acids, phosphate-containing compounds, nitrogen components, and nucleic acids (Fig. S1 and S2 and Table S5). Strikingly, the whole biosynthetic pathways for tryptophan and histidine were missing in the pathogens (Fig. 6). The majority of the biosynthetic genes are clustered together as different operons on the ancestral *L. crescens* genome; however, a complete deletion of several related operons and other genes highlights the dependency of these intracellular pathogens on receiving these nutrients from their hosts.

For the genes that were gained in the pathogens, we found that they encode multiple transporters, enzymes involved in DNA/RNA synthesis, regulation, or small molecule metabolism, and several transmembrane proteins (Table S4). Some of them might contribute to the pathogen's ability to adapt and proliferate in the intracellular environment of host cells. For example, the survival protein SurE is a metal-dependent nucleotide phosphatase that has been shown to be essential for bacterial pathogenesis and survival in the stationary phase and in harsh conditions (45). Further, the multiple transporters might play important roles in exchanging chemical components between the pathogens and the plant host (46). Thus, both the gene-loss and gene-gain events that happened in the ancestor of *Liberibacter* pathogens seem to have defined the molecular foundation of their endophytic habitats.

**Extensive sequence and structure analyses identify potential virulence factors, including several polymorphic toxins.** Since pathogenicity is an acquired trait for *Liberibacter* bacteria, we hypothesized that such ability is attributed to the unique pathogenicity-related genes that were gained by the ancestor of these pathogens during evolution (or gene-gain events). Therefore, the potential virulence factors, such as toxins and effectors, should be among the gene-gain list that we identified (Table S4). However, of the proteins on our list, almost half of them are hypothetical proteins with no functional annotation. To accurately uncover the function of these proteins, we conducted a series of analyses to examine the sequence/structural features of these proteins, dissect their domain components, establish the distant relationship between these domains with the known Pfam domains, and synthesize the function of the proteins by combining the domain annotations. This systematic analysis has allowed us to identify three groups of proteins as potential toxins/effectors (Table S4).

**FIG 5** Legend (Continued)

gained or lost during evolution according to the phyletic patterns (presence or absence) of orthologous genes in both ancestral free-living *L. crescens* and pathogenic decedents. (B to D) Phylogenetic verification of gene transfer events of three gene products, exonuclease-endonuclease-phosphatase (EEP), peptidase M75 (iron transporter), and sodium:dicarboxylate symporter family (SDF) protein, to the pathogenic ancestor. The trees were inferred using maximum likelihood (ML) method, and the supporting values from 100 bootstraps are shown for major branches. Sequences were annotated by their species and strains, colored accordingly. Some sequences from different "*Ca.* Liberibacter asiaticus" strains are identical and share the same NCBI accession number, so we use one sequence to represent them, indicated by an asterisk (*). Specifically, "*Ca.* Liberibacter asiaticus"* in panel B indicates sequences from "*Ca.* Liberibacter asiaticus" strain psy62, strain gxpsy, strain A4, strain AHCA1, strain Ishi-1, and strain JXGC, "*Ca.* Liberibacter asiaticus"* in panel C indicates sequences from strain A4, strain AHCA1, strain Ishi-1, and strain JXGC, and "*Ca.* Liberibacter asiaticus"* in panel D indicates strain psy62, strain gxpsy, strain A4, strain AHCA1, strain Ishi-1, and strain JXGC.

# A Tryptophan biosynthesis pathway

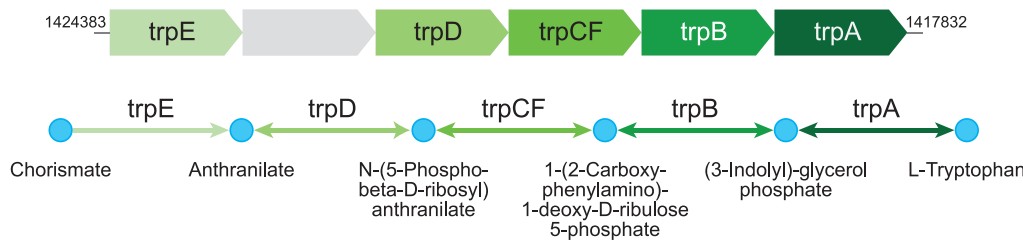

# B Histidine biosynthesis pathway

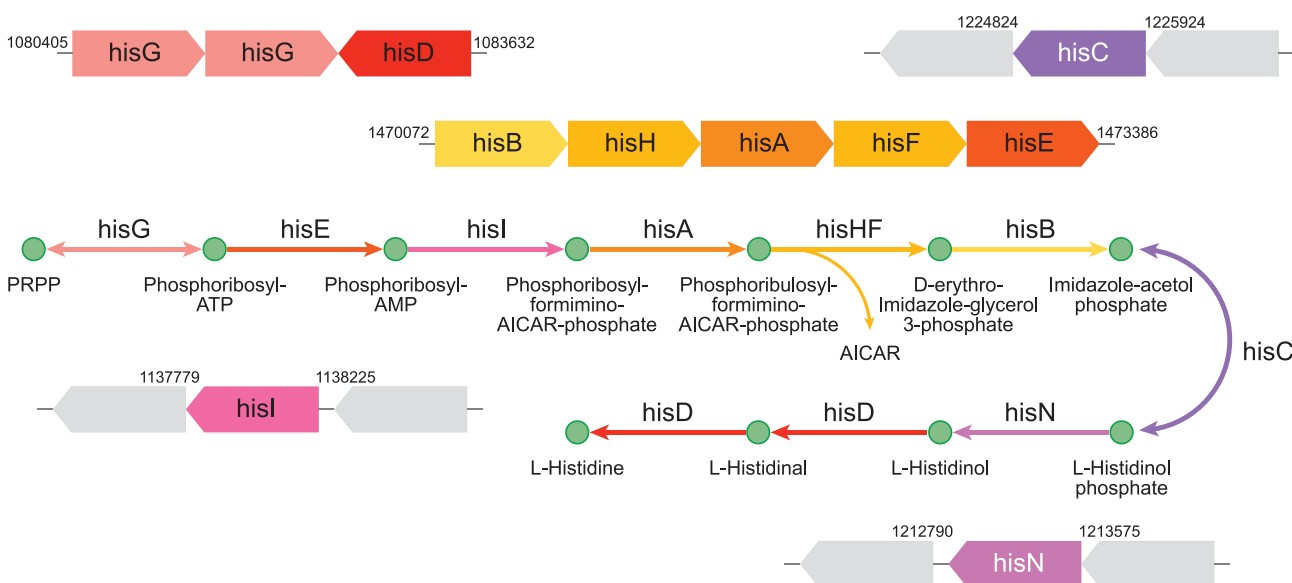

**FIG 6** KEGG pathway analysis reveals a complete deletion of a chain of enzymes of the pathways responsible for biosynthesis of tryptophan (A) and histidine (B) in the pathogenic *Liberibacter* species. In each case, operonic structure of biosynthesis enzymes in the ancestral *L. crescens* BT-1 genome is presented above, with the enzyme genes in colored arrow blocks and gene boundaries indicated by genomic locations; the KEGG pathway is shown below, with reaction steps in arrowed lines, colored according to the operonic blocks. The gray blocks indicate the genes that are not associated with the pathway and not lost in pathogenic *Liberibacter* species.

The first toxin group comprises two hypothetical proteins (e.g., WP_012778667.1 and WP_012778668.1 in "Ca. Liberibacter asiaticus" strain gxpsy). By sequence analysis, we found that these two proteins share a unique domain architecture containing two previously unrecognized domains, a PD domain (designated for the conservation of two residues, Pro and Asp; Fig. S3) and a Ntox52 domain (Fig. S4). No function has been associated with these domains. However, by studying the proteins containing either the PD or Ntox52 domains, we found that they are also coupled with other domains that are components of polymorphic toxin systems (PTSs). PTSs are a vast class of toxin systems that we have recently identified in bacteria and archaea (47–50). They serve as primary weaponry for many bacterial pathogens (51–54) which were exported through different secretion pathways, such as T2SS, T5SS, T6SS, and T7SS (47, 48, 50, 55, 56). Despite the extensive diversity of these polymorphic toxins, we were able to dissect the underlying principles of these proteins: (i) the toxins typically contain multiple modular architectures with N-terminal secretion-related domains, central repeats or linker domains, pretoxin domains, and C-terminal toxin domains, and (ii) the toxins display a tremendous polymorphism in domain composition as they diversify through domain recombination or shuffling (47, 48, 50). Therefore, the association of toxin-related domains is the most prominent feature of toxin proteins.

We found that both the PD and Ntox52 domains display a typical association with other toxin-specific domains (Fig. 7A). The PD domain is frequently coupled with long

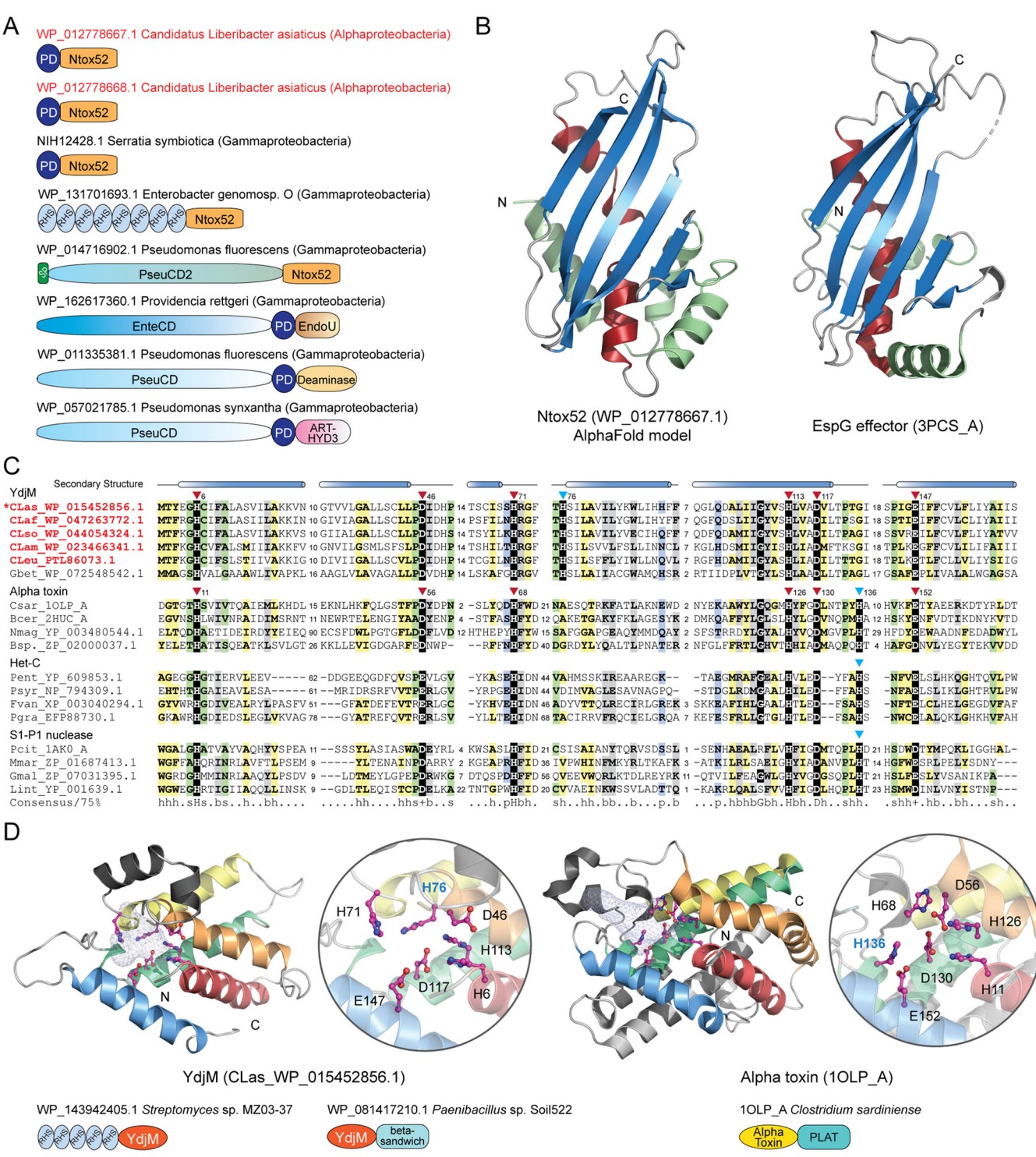

**FIG 7** Identification of potential protein toxins. (A) Domain architectures of representative polymorphic toxins containing the PD domain and the Ntox52 domain. The PD toxins of "*Ca.* Liberibacter asiaticus" are highlighted in red. Domain architectures are labeled by accession numbers, species names, and their lineages in parentheses. Domain architectures are not drawn to scale. (B) The AlphaFold2 model of Ntox52 and its structural similarity with the EspG effector (PDB: 3PCS_A). (C) Multiple sequence alignment between the YdjM, alpha toxin, Het-C, and S1-P1 nuclease domains. The conserved catalytic residues are highlighted with a black background. (D) Structural comparison of YdjM and the alpha toxin domains. Representative domain architectures of these toxin proteins are shown.

N-terminal regions and several C-terminal toxin domains such as the EndoU nuclease, deaminase, and ADP-ribosyltransferase (ART-HYD3) domains (Fig. 7A), which were identified in our earlier studies (47, 48, 50, 57). The Ntox52 domain, on the other hand, is always located at the C terminus of toxin proteins, a position that the toxin domain typically occupies, and coupled

with other toxin-specific central modules, such as RHS (rearrangement hot spot) and PseudoCD2. These lines of evidence strongly suggest that the two identified proteins represent novel polymorphic toxins where the PD domain is a pretoxin module and the Ntox52 is a toxin module. As no homolog was found in profile-profile sequence searches, we next modeled the structure of Ntox52 using AlphaFold2, a recently developed method that can predict the protein structures with atomic accuracy even where no similar structure is known (58). With the structure model of Ntox52, we were able to identify its distant homologs by using structure similarity searches implicated in the DALI program (59). These analyses revealed that the Ntox52 domain adopts an alpha plus beta structure (Fig. 7B) which shows significant similarity to several known effectors, including enteropathogenic *E. coli* EspG effector (PDB: 3PCS_A; DALI Z-score = 7.5; Fig. 7B) (60), *Shigella flexneri* VirA effector (PDB: 4FMC_E; DALI Z-score = 7.5) (61), and nematode CPI effector (PDB: 4IT7_D; DALI Z-score = 5.0) (62). These effectors have been demonstrated to function as an inhibitor of enzymes, including GTPase and peptidase, suggesting that these structures define a class of new enzyme inhibitors. Given that Ntox52 is mostly similar to the EspG/VirA effectors (Fig. 7B), we propose that Ntox52 might function as an inhibitor of a host enzyme, most likely a GTPase.

Another potential toxin is the YdjM protein (Fig. 7C). Our profile-profile comparison revealed that YdjM is related to several toxin domains, including bacterial alpha toxin (probability 95% of profile-profile match), a membrane-disrupting phospholipase responsible for gas gangrene and myonecrosis in *Clostridium perfringens*-infected tissues (63), and the HET-C domain (probability 93% of profile-profile match), which is also a toxin component that we identified in bacterial PTSs and in the fungal nonallelic incompatibility system (48, 64) (Fig. 7C). Although they share a low sequence identity, these domains display conservation in both the structural composition and the catalytic core (Fig. 7C and D). The catalytic core of the known alpha toxin and Het-C domains involves seven conserved residues, six of which are preserved in the YdjM domain. However, our structural modeling suggested that the YdjM-specific His76 serves an equivalent role as His136 of the alpha toxin (Fig. 7D). Further, by analyzing the domain architectures of many other proteins containing the YdjM domain, we found that (i) several bacterial RHS-type toxins use YdjM as their C-terminal toxin module (e.g., WP_143945134) (Fig. 7D) and (ii) the YdjM domain is predominantly coupled with a novel beta-sandwich domain (Fig. 7C and Fig. S5), like the alpha toxin, which has a beta-sandwich domain to facilitate toxin localization at the membrane (63). Thus, based on these lines of evidence, we proposed that YdjM is a novel phospholipase toxin which might disrupt the membrane of host cells.

The third toxin group includes several proteins belonging to the EEP (endonuclease/exonuclease/phosphatase) family (Fig. 5B and Table S5). They contain an N-terminal signal peptide, indicating that they are secreted and therefore might target host cells. The EEP family comprises many enzymes with different activities, from DNA I-like nucleases, endonucleases of retrotransposons, and inositol polyphosphatases to phosphodiesterases (Pfam ID: PF03372). By a phylogenetic analysis of the *Liberibacter* EEP-related sequences and other known EEP enzymes (Fig. S7), we found that the *Liberibacter* EEP and related sequences form a distinct clade which is more closely related to a clade of DNA I-like nucleases, which also include *Haemophilus ducreyi* cytolethal distending toxins (1SR4_B) (65) and *Salmonella* typhoid toxin (4K6L_F) (66). Therefore, it is possible that the secreted *Liberibacter* EEP proteins act as nuclease toxins targeting the host DNAs.

## DISCUSSION

HLB is the most destructive disease of citrus worldwide and is associated with several species of *Candidatus* Liberibacter, a psyllid-transmitted, phloem-limited alpha proteobacteria. Despite intense scrutiny, molecular mechanisms of the HLB pathogenesis remain to be elucidated. The fact that these pathogens are obligate intracellular bacteria and are not culturable *in vitro* has made experimental, functional studies extremely challenging. Thus, computational mining of available genome information has become a promising strategy to uncover potential pathogenesis mechanisms of HLB and to guide the directed experimental studies. Here, we present an in-depth comparative genomic analysis of several *Liberibacter* species, including those pathogens associated with citrus HLB and potato ZC, along with

the ancestral, nonpathogenic *Liberibacter* species, *L. crescens*. We successfully identified the major genomic changes that may contribute to the establishment of endophytic habitats and pathogenicity of the *Liberibacter* pathogens.

One of the major genome variations found by our comparative genomic analysis resides at the prophage loci (Fig. 2 and 3). Prophages in *Liberibacter* have been studied extensively in the past, but the results are controversial. On the one hand, the primary HLB pathogen, "*Ca.* Liberibacter asiaticus," is found to carry two prophage loci whose lytic cycle was activated during bacterial infection in plant (41), indicating that the prophage might be involved in pathogenicity or infection. On the other hand, the absence of the prophage in a Japanese strain of "*Ca.* Liberibacter asiaticus" Ishi-1, which also induces severe symptoms in citrus, suggests that the prophage might be dispensable for pathogenicity. Further, many different prophage loci are identified in both *Liberibacter* pathogens and the free-living ancestor (36, 42), making it more difficult to understand their functional contribution. To resolve these puzzles, we conducted a unique, comprehensive operonic association analysis to recover all the prophage loci in the *Liberibacter* genomes and systematically annotate the conserved phage components. According to both the prophage composition and phylogeny of the conserved terminase, we were able to classify these diverse prophage loci into four major types, namely, LC1, LC2, SC, and UT. This result has clarified several points of confusion about the presence of the prophage loci in the *Liberibacter* species. For example, our phylogenetic analysis has linked the recently classified type 4 prophages from the *Liberibacter* pathogens (42) to the earlier LC2 prophage from the ancestral *L. crescens* (36). We also uncovered a previously undetected prophage type, the UT phage, that is present in all the "*Ca.* Liberibacter asiaticus" strains. Further, we were able to identify the LC2 and UT prophages in the Japanese strain which was previously thought to contain no prophage loci (27). Thus, our detailed annotation and classification of prophages have provided a genomic resource for studying the prophages in the *Liberibacter* bacteria.

Importantly, our analysis revealed that these different types of prophages originated through multiple independent gene transfer events to *Liberibacter* at different evolutionary stages (Fig. 4C). However, the SC-type prophage is the only type that was acquired by the ancestor of the pathogens. Thus, this evolutionary association suggests that the acquisition of the SC phage might contribute to the emergence of pathogenicity. Indeed, many bacterial toxins are known to be carried by prophage loci (38–40). On average, a typical SC prophage locus contains about 35 genes. While the majority of these genes encode structural components and enzymes involved in phage replication and transcription, others remain uncharacterized. Therefore, a systematic analysis of the SC prophage components will be necessary to understand their role in the *Liberibacter* pathogens. We also observed an unusual divergence between the SC prophage loci from different *Liberibacter* pathogens and different strains of the same pathogen (Fig. 4B and C), distinct from other genomic regions and other types of prophages. This rapid evolutionary pattern of the SC prophage suggests that some of its components may have acted at the interface of host-pathogen interactions and gained the upper hand during the evolutionary arms race (67, 68).

Our comparative genomic analysis also identified genomic (nonprophage) genes that were lost or gained at the ancestor of the *Liberibacter* pathogens. To be noted, of the 323 orthogroups (335 genes) that were lost in all the pathogens, 56 genes were also identified as the culture-essential genes for *L. crescens* in a Tn5 transposon mutagenesis screening experiment (69). Functional and pathway analysis revealed that biosynthetic pathways of several essential amino acids in the pathogens were disrupted by gene loss, consistent with the earlier studies (36, 70). Some of the genes that were gained in the ancestor of the pathogens encode transporters or transmembrane proteins, which may also contribute to the molecular adaptation of the bacteria to endophytic habitats. Further functional characterization of the genes on the gene-loss and gene-gain lists holds promises for advancing our understanding about the nutrient dependence of endophytic *Liberibacter* bacteria and enabling rational design of new strategies to culture them *in vitro*.

More importantly, our detailed sequence and structural analyses have identified several potential protein toxins. Protein toxins or effectors are the major pathogenicity

components of bacteria-caused diseases. However, the toxins that are responsible for HLB and other related diseases have remained enigmatic for years. Early genome annotation did not reveal any bacterial pathogenesis-related genes in the genome. Although several studies have attempted to examine the genes encoding small, secreted proteins, candidates thus far are limited to only certain "*Ca.* Liberibacter asiaticus" strains (19–21, 23–25, 71). We had also tried to identify toxins in *Liberibacter* by utilizing our previously established toxin domain profiles of polymorphic toxin systems (47, 48, 50); unfortunately, no candidate was identified. This suggests that the *Liberibacter* pathogens might utilize some novel toxins whose toxin domains are not yet identified. Indeed, by studying the hypothetical proteins that were found to follow a gene gain pattern, we were able to uncover three new toxin groups, including Ntox52 toxins, a YdjM phospholipase toxin, and a secreted EEP protein. Despite the fact that they were previously uncharacterized, we show that (i) they are closely related to several known effector families and (ii) their toxin domains are fused with other typical domains of the polymorphic toxins. These strongly support their role as toxins in the *Liberibacter* pathogens. In addition to the above toxin candidates, other hypothetical proteins in the gene-gain list may also be potential toxins. One such example is the protein (WP_015452959.1) that was recently found to cause cell death in the leaves of *Nicotiana benthamiana* (71).

To be noted, only the EEP toxin and several other proteins that were gained during evolution harbor the signal peptide, the export signal of T2SS (48), or the transmembrane (TM) region, which directs the protein localization at the bacterial cell membrane so that their coupled toxin module can be exported. However, two other toxin groups, Ntox52 and YdjM, do not contain signal peptide, TM, or other secretion-related domains (48). It is possible that *Liberibacter* might utilize a yet unknown secretion pathway to export these toxins. Alternatively, these toxins might be exported when these *Liberibacter* pathogens undergo phage lytic cycle. It has been reported that prophages of the SC type, the one acquired by the ancestor of the pathogens, are activated and convert from lysogenic cycle to lytic cycle when pathogenic *Liberibacter* bacteria infect the host plant (41). The lytic cycle of prophages will result in the destruction of bacterial cell membrane, which could release their cytoplasmic molecules, including toxins or effectors. Disruption of bacterial cells to release toxins or effectors has been used by bacteria in the situation of kin selection (72). In the so-called bacteriocin system, the bacteria produce cytoplasmic toxin protein, immunity protein, and lysis protein, among which the lysis protein would lyse bacteria to facilitate the release of both the toxin and immunity proteins to promote self-nonself recognition in the bacterial community (72).

This research expands our recent efforts in using computational means to dissect the molecular mechanisms of complex organismal interactions (47, 48, 50, 73). Organismal or species interactions, interspecific, intraspecific, or between pathogen and host, are the mainstay of life (48). Driven by the evolutionary arms race, the proteins mediating these interactions, such as toxins, effectors, or virulence factors, typically evolve rapidly and are difficult to identify using traditional experimental and computational methods (74). The major contribution of our research in this regard is to use dedicated protein domain-centric analysis strategies, comparative genomics, and evolutionary theory to identify such toxin/effector components and to formulate the principle of the systems behind these interactions (47, 48, 73, 74). This approach has led to the discovery of several distinct classes of conflict systems, including bacterial polymorphic toxin systems involved in kin selection and bacteria-host interactions (47, 48, 50), Crinkler-RHS (CR) effector systems at the interface of eukaryotic pathogen/symbiont-host interactions (73), nucleotide-centric conflict systems (75), DNA modification systems deployed in phage-bacteria interactions (76), and viral pathogenicity factors involved in coronavirus-host interactions (67, 68). Many of our computational predictions in protein function and the organizational principles of the systems have facilitated the creation of new concepts and directions in several research fields and have been later experimentally validated, including the recent discoveries of the enzymatic function of animal and plant Toll/interleukin-1 receptor (TIR) proteins (77, 78), adenosine methylation enzymes (79, 80), type III CRISPR-Cas systems (81, 82), and novel immunoglobulin proteins and ion channel proteins in SARS-CoV-2 (67, 68), in addition to much work on PTSs (47, 48, 50).

As an extension to understand the interactions between bacterial pathogens and their host, we present our identification of the elusive pathogenicity factors in *Liberibacter* bacteria. This knowledge will open doors to developing and deploying new strategies for the detection of *Liberibacter* pathogens and treatment of the associated diseases. Both the SC prophage loci and the genes gained in the ancestor of the pathogenic strains, especially the newly identified toxin groups, can serve as biomarkers for specific detection of the pathogens. The association of these genes with pathogenic species suggests that some of them may play a role in pathogenesis. This hypothesis can be tested directly using targeted, functional experiments. We envision that future research informed and enabled by our genomic analysis holds great promises to elucidate the pathogenesis mechanisms of *Liberibacter* bacteria, leading to long-sought solutions for curing HLB and other related diseases.

## MATERIALS AND METHODS

**Genomes that were investigated in this study.** A total of 12 genomes of *Liberibacter* species were collected from the nucleotide database of the National Center of Biotechnology Information (NCBI). Their genome annotations including the coding sequences (CDS) and RNA genes were extracted from NCBI GenBank files. The encoded protein sequences were retrieved from the NCBI protein database. The detailed information of the genomes used in this study can be found in Table S1.

**Pairwise genome comparisons.** To systemically identify the genomic features and variations among *Liberibacter* species, we performed pairwise whole-genome comparisons by using the local TBLASTX (83) program with the cutoff E value of 0.001 serving as the significant threshold. The results were visualized using Artemis Comparison Tool (ACT) (84) with a cutoff matching sequence length at 500 bp and modified using Adobe Illustrator. The "*Ca.* Liberibacter europaeus" genome has 15 contigs, and in order to compare it to other genomes, we linked these contigs based on the following order of their accession numbers: PSQJ01000001.1, PSQJ01000002.1, PSQJ01000004.1, PSQJ01000005.1, PSQJ01000006.1, PSQJ01000007.1, PSQJ01000008.1, PSQJ01000009.1, PSQJ01000010.1, PSQJ01000011.1, PSQJ01000012.1, PSQJ01000013.1, PSQJ01000014.1, PSQJ01000015.1, and PSQJ01000003.1.

**Protein sequence search and analysis.** To collect protein homologs, iterative sequence-profile searches were conducted using the Position-Specific Iterated BLAST (PSI-BLAST) program (83) against the nonredundant (nr) protein database of NCBI with a cutoff E value of 0.005 serving as the significance threshold. Similarity-based clustering was performed by BLASTCLUST, a BLAST score-based single-linkage clustering method (https://ftp.ncbi.nih.gov/blast/documents/blastclust.html). Multiple sequence alignments (MSA) were built by the KALIGN (85), MUSCLE (86), or PROMALS3D (87) programs, followed by careful manual adjustments based on the profile-profile alignment and the secondary structure information generated by the JPRED program (88). A consensus method was used to calculate the conservation pattern of the MSA based on different categories of amino acid physicochemical properties developed by Taylor in 1986 (89). Consensus was calculated by examining each column of the MSA to determine whether a threshold fraction (either 75% or 80%) of the amino acids belongs to a defined category. Then, the MSA was colored using the CHROMA program (90) based on the calculated consensus sequence and further modified using Adobe Illustrator or Microsoft Word. The HHsearch program was used for profile-profile comparison (91). Signal peptide and transmembrane region prediction was detected using the Phobius program (92). Potential open reading frames were detected using the ORFfinder program (https://www.ncbi.nlm.nih.gov/orffinder/).

**Molecular phylogenetic analysis.** We used both the maximum likelihood (ML) analysis, implicated in the MEGA7 programs (93), and Bayesian inference (BI), implemented in the BEAST 1.8.4 program (94), to reconstruct the phylogenetic relationships of genes and proteins.

To infer the phylogeny of *Liberibacter* species, we selected three genes, including 16S rRNA, 23S rRNA, and DNA polymerase I, collected their homologs from *Liberibacter* bacteria and other related sequences using BLASTN or BLASTP programs, and generated MSAs which were further analyzed using the ML analysis. A GTR (generalised time reversible) model was applied to the RNA sequences, and a JTT (Jones-Taylor-Thornton) model was applied to the protein sequences. Initial tree(s) for the heuristic search were obtained automatically by applying neighbor-join and BioNJ algorithms to a matrix of pairwise distances estimated using the maximum composite likelihood (MCL) approach and then selecting the topology with a superior log-likelihood value. For both data types, a discrete Gamma distribution was used to model evolutionary rate differences among sites (four categories). A bootstrap analysis with 100 repetitions was performed to assess the significance of the phylogenetic grouping.

To reconstruct the evolutionary history of terminases, we first applied the ML analysis in which a WAG (Whelan and Goldman) model with a discrete gamma distribution (four categories) was used to model rate heterogeneity among sites. A bootstrap analysis with 100 repetitions was performed to assess the significance of the phylogenetic grouping. We also applied the BI analysis with a JTT model and a discrete Gamma distribution (four categories). Markov chain Monte Carlo (MCMC) duplicate runs of 10 million states each, sampling every 10,000 steps, was computed. Logs of MCMC runs were examined using Tracer 1.7.1 program (94). Burn-ins were set to be 2% of iterations.

All trees with the highest log-likelihood from the ML analysis were visualized using the MEGA7 program (93). The tree is drawn to scale, with branch lengths measured in the number of substitutions per site. The bootstrapping values from ML analysis and/or the posterior values from Bayesian inference analysis are shown next to the branches.

**Gene neighborhood analysis of the prophage loci.** In order to identify the prophage loci, we utilized the gene neighborhood analysis. Unlike many earlier studies (30, 36, 42), which relied on the existing phage database and sequences, our analysis was based on the operonic association to comply with the extreme diversity of prophage loci. As the terminase is the core component of prophage, we used it as a marker to identify potential prophage loci on the genome. We collected the upstream and downstream gene neighbors of the terminase from the NCBI GenBank files. All protein sequences were clustered using the similarity-based BLASTCLUST program (https://ftp.ncbi.nih.gov/blast/documents/blastclust.html). Protein clusters were further annotated with the conserved domains which are identified by the hmmscan program searching against Pfam (91, 95) and our own curated domain profiles.

**Identification of the genes gained or lost at the ancestor of pathogenic species.** To identify the genes that were gained or lost at the ancestor of all *Liberibacter* pathogens, we first utilized the OrthoFinder v2.2.3 program (44, 96) with default parameters to infer groups of orthologous gene clusters (orthogroups) among all 12 *Liberibacter* proteomes based on protein homology detection by Diamond (97) and MCL clustering (98). Orthogroups comprise genes of related species that evolved from a common gene ancestor by speciation (99). Given the common ancestry of all *Liberibacter* species, the majority of the genes should share common ancestors and be preserved in all these species, whereas the genes that were gained or lost at the ancestor of the *Liberibacter* pathogens will display a different presence in either pathogenic *Liberibacter* species or nonpathogenic ancestor. Therefore, we extracted these orthogroups using a custom Python script based on the following criteria. (i) Orthogroups with gene loss: the ones present in nonpathogenic *L. crescens* but not in any of the pathogenic "*Candidatus* Liberibacter" species. (ii) Orthogroups with gene gain: the ones that are found in at least 4 "*Candidatus* Liberibacter" species (a total of 5 species used in this study) but not in *L. crescens* genomes.

Importantly, by conducting case-by-case phylogenetic analyses, we realized that there were still some false positive and false negatives in the ortholog clustering result. To overcome this methodological limitation, we utilized another strategy given the special situation of *Liberibacter* species. For ortholog clustering, the gene transfers between bacterial genomes are the major obstacle. In the case of *Liberibacter* bacteria, all pathogenic species are endophytic bacteria and their life cycle is entirely restricted in the host cells. Therefore, they have little chance to exchange genes with other bacteria, evolution of their genes should have been influenced mainly by sequence diversification, and their relationship should be readily computationally tractable using pairwise sequence similarity scores. Based on this evidence, we conducted genome-wide BLASTP searches against the NCBI-NR database using all "*Ca.* Liberibacter asiaticus" proteins as queries. The results showed a good correlation between sequence similarity scores and the evolutionary distance. Thus, by this simple BLAST search, we were able to identify (i) the proteins in the nonpathogenic ancestor that have no close homologs in pathogenic *Liberibacter* species and (ii) the proteins in pathogenic species that have no close homologs in the ancestor.

To validate the above results on gene gain and gene loss, we carried out extensive bootstrapped maximum likelihood phylogenetic analyses using the MEGA7 program for all the identified orthogroups by including both *Liberibacter* proteins and other related homologs that were identified from the NCBI-NR database. These three steps led to an identification of 323 orthogroups (about 335 genes) that are found only in nonpathogenic *L. crescens* and 35 orthogroups that are present only in pathogenic bacteria (Table S3 and S4). The number of gene-loss and gene-gain genes might differ between species or strains due to genome-specific duplications.

**Protein function analysis.** We used three levels of analysis to conduct the functional annotation of the proteins that were identified in this study. First, we retrieved the annotation information from the NCBI RefSeq database and Gene Ontology (GO) terms using the Blast2GO program (100). Their involvement in the molecular synthesis pathways was determined by mapping the KEGG database (101). Second, for those proteins that have no functional annotation, we annotated them using the conserved domains which were identified using the hmmscan program searching against Pfam (91, 95) and our own curated domain profiles. Finally, for proteins with potential uncharacterized domains, we conducted detailed case-by-case analysis of protein sequences and structures, such as homologous sequence searches (PSI-BLAST) (83), multiple sequence alignment analysis (KALIGN, MUSCLE, PROMALS3D) (85–87), secondary structure prediction (JPRED) (88), and sequence-profile/profile-profile searches (HHsearch) (91), to dissect their domain architectures, identify the conserved sequence/structural features, and predict aspects of their biochemical and biological function.

**Protein structure prediction and analysis.** The MODELLER (version 9v1) program (102) was utilized for homology modeling of the tertiary structure of the "*Ca.* Liberibacter asiaticus" YdjM (WP_015452856.1) by using the alpha toxin (1OLP_A) as a template. The sequence identity between the template and the targets is 9%. Since in these low sequence-identity cases sequence alignment is the most important factor affecting the quality of model (103), the alignment used in this analysis has been carefully built and cross-validated based on the information from HHsearch and edited manually using the secondary structure information. For the Ntox52 domain, as no distant homolog can be identified, we applied the recently developed deep learning system AlphaFold (58) to predict the structure of a representative toxin protein in "*Ca.* Liberibacter asiaticus" (WP_012778667.1). According to the benchmark, AlphaFold generates the structure with atomic accuracy even where no similar structure is known (58). To identify the distantly related homologs of Ntox52, we conducted structure similarity searches via the DALI program, which generates an optimal pairwise structural alignment based on the similarity of local patterns extracted from contact maps (59). Other structural analysis and comparison operations were conducted using the molecular visualization program PyMOL (104).

**Data availability.** All relevant data are included in the article and/or the supplemental material.

## SUPPLEMENTAL MATERIAL

Supplemental material is available online only.

**SUPPLEMENTAL FILE 1**, PDF file, 1.3 MB.

**SUPPLEMENTAL FILE 2**, XLSX file, 0.04 MB.

**SUPPLEMENTAL FILE 3**, XLSX file, 0.02 MB.
**SUPPLEMENTAL FILE 4**, XLSX file, 0.1 MB.
**SUPPLEMENTAL FILE 5**, XLSX file, 0.4 MB.

## ACKNOWLEDGMENTS

Y.T., C.W., T.S., H.L., and D.Z. are supported by the Saint Louis University Start-up Fund. T.-F.H. is supported by the National Institute of Food and Agriculture Hatch Project 02413. X.W. is supported by a USDA National Institute of Food and Agriculture Hatch project 1018100, National Science Foundation EPSCoR RII Track-4 Research Fellowship (NSF OIA 1928770), an Alabama Agriculture Experiment Station ARES Agriculture Research Enhancement, Exploration and Development (AgR-SEED) award, as well as a generous laboratory start-up fund from Auburn University College of Veterinary. X.L. is supported by the National Institute of Food and Agriculture Hatch Project 02634. D.Z. and K.D.S.G. are supported by the USDA/ARS − State Partnership FY 2021 Potato Research Program.

D.Z. conceived the overall project. Y.T., C.W., T.S., and H.L. performed the phylogenetic analysis. Y.T and D.Z. performed genome comparisons, GO and pathway analysis, protein domain identification, domain architecture analysis, and structure analysis/modeling. Y.T. and C.W. conducted the operonic analysis of prophages and studied the function of phage components. Y.T., C.W., and R.F.D.S. were involved in ortholog analysis. Y.T. and R.F.D.S. contributed to the data organization. Y.T., X.T., K.D.S.G., T.F.H., X.W., X.L., and D.Z. contributed to the design of the analysis strategies. Y.T., H.L., X.T., K.D.S.G., T.F.H., X.W., X.L., and D.Z. interpreted the results. Y.T. and D.Z. wrote the manuscript. All authors have read, revised, and commented on the manuscript.

We declare no conflicts of interest.

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
