## [Reviewer comments · Microbiology Spectrum]

**Microbiology
Spectrum**

Comparative phylogenomic analysis reveals evolutionary genomic changes and novel toxin families in endophytic *Liberibacter* pathogens

Yongjun Tan, Cindy Wang, Theresa Schneider, Huan Li, Robson de Souza, XM Tang, Kylie Swisher Grimm, Tzung-Fu Hsieh, Xu Wang, Xu Li, and Dapeng Zhang

Corresponding Author(s): Dapeng Zhang, Saint Louis University

Review Timeline:

Submission Date:	June 8, 2021
Editorial Decision:	July 9, 2021
Revision Received:	August 10, 2021
Accepted:	August 10, 2021

Editor: Lindsey Burbank

Reviewer(s): Disclosure of reviewer identity is with reference to reviewer comments included in decision letter(s). The following individuals involved in review of your submission have agreed to reveal their identity: David Botero (Reviewer #1); Jeannie M. Klein-Gordon (Reviewer #3)

Transaction Report:

DOI: <https://doi.org/10.1128/Spectrum.00509-21>

July 9, 2021

Dr. Dapeng Zhang
Saint Louis University
Biology
St. Louis, MO 63103

Re: Spectrum00509-21 (Comparative phylogenomic analysis reveals evolutionary genomic changes and novel toxin families in endophytic *Liberibacter* pathogens)

Dear Dr. Dapeng Zhang:

The reviewers highlighted several strengths of your manuscript and the topic is very important in the context of plant diseases in agriculture. However, there were some concerns that need to be addressed. In particular, be sure to put your new findings into the context of similar studies that have previously been published and accentuate the novel findings of this study. Please also address the other comments of the reviewers with attention to clarity of the writing.

Thank you for submitting your manuscript to Microbiology Spectrum. When submitting the revised version of your paper, please provide (1) point-by-point responses to the issues raised by the reviewers as file type "Response to Reviewers," not in your cover letter, and (2) a PDF file that indicates the changes from the original submission (by highlighting or underlining the changes) as file type "Marked Up Manuscript - For Review Only". Please use this link to submit your revised manuscript - we strongly recommend that you submit your paper within the next 60 days or reach out to me. Detailed information on submitting your revised paper are below.

Link Not Available

Sincerely,

Lindsey Burbank

Journals Department
Reviewer comments:

Reviewer #1 (Comments for the Author):

The authors present novel insights into pathogenicity gene markers of *C. Liberibacter* genus. Those genes will be a main source in pathogenicity studies related to this genus. They go far beyond previous studies in the field. They also found several gene operons lost at the metabolic and other molecular levels. The phylogenetic and genomic analyses show interesting insights into the evolutionary pathway of *C. Liberibacter* and the main differences between pathogenic and non-pathogenic bacteria. Finally, the authors test and validated their hypothesis about host-live style of *C. Liberibacter* species.

Minor commentaries below:

L57. I am not sure about this affirmation. The work in effectors and genome analyses performed by Coaker et al., 2019 "Genome-wide analyses of *Liberibacter* species provides insights into evolution, phylogenetic relationships, and virulence factors" point out to several effectors implied in pathogenicity. Surely, the previous study is not so broad and deep as the present study present here but made an important contribution in the past.

L94-104. Maybe I am wrong... Again, I recommend to review the same research of Coaker and figure out if all the affirmations are still true, for example, L98. "these proteins are mainly present in one strain of the *C.L. asiaticus* species, and not found in other *C.L. asiaticus* strains ...". As I understand, Coaker et al. (2019) used several strains of CLas for the effector's prediction and comparisons.

L129. Reference 31. So, from that research, only the sequence of papaya was used?

L407. Include ref 31 and verify if was also in small proteins.

L565. Please attach curated domain profiles.

Reviewer #2 (Comments for the Author):

This manuscript falls in the category of "re-analyses of large datasets that provide additional insights". The authors present a comprehensive comparative genomic analysis between *Candidatus Liberibacter* genomes both pathogenic and non-pathogenic. Although much of the information in this manuscript might not be novel, the authors present several novel findings that are significant for understanding how *Liberibacter* gains its pathogenicity features and thus based on this the manuscript has its merits. Also, the authors present an overview of the major distinctive features between the genomes of pathogenic and non-pathogenic species, and specifically characterize toxins and prophages, of which both are distinctive for pathogenic species. The section discussing toxins and comparing these toxin classes to other species is very interesting. The manuscript presents a wealth of data, much of it is in the form of supplementary files. The authors overlook some of the important genome features and this needs to be addressed (see below).

As, several comparative genomics studies have already been published, it would have been interesting whether data in this study is congruent with previous findings. For instance, whether the gene lost in the pathogenic *Liberibacter* genomes are similar to results from previous studies (such as in references 59 and 60).

Also, some of the results section are unnecessarily long and very descriptive, it would be better to be succinct in presenting the findings and thus more information can be presented. For instance, the first five sections in the results are very descriptive and get be significantly shortened. The authors claim recombination and inversions are prominent, but do not explain how and based on what. A lot of the claims are not backed up with the data here.

The authors discuss recombination as one of the forces driving evolutionary changes whereas as a matter of fact, it is horizontal gene transfer, due to integration of these diverse prophages. The authors do not mention HGT which is a major evolutionary force in fact.

Furthermore, authors do not present any data or mention proteins with signal peptides in the genes that have been acquired in the pathogenic species, and do not discuss any of the secretions systems as potentially part of the acquired genes/clusters driving evolutionary changes in the pathogenic species.

Furthermore, the UT and ZC1 prophages, are two prophages identified here but not well discussed. Is there anything significant about these prophages?

Please check the English for grammar mistakes, avoid using "it" as an anticipatory pronoun, it can be confusing. L81: what is in a similar vein? Please fix. Some of the introduction, L97-L102 is somewhat vague and needs some further work.

Note, please double check the references. For instance reference 61 is erroneously cited in Line 416-417. I have not checked all the references, this is something the authors will have to do.

Some specific inquiries:

L191: fragments of what? Please re-write.

L208-210: can you elaborate here on this topic, it is not evident.

Fig 4C is not cited in the manuscript.

Reviewer #3 (Comments for the Author):

The manuscript entitled "Comparative phylogenomic analysis reveals evolutionary genomic changes and novel toxin families in endophytic *Liberibacter* pathogens" by Tan et al. identifies prophage loci and toxin-associated genes within *Liberibacter* genomes as a means to understand the emergence of endophytic, pathogenic *Liberibacter* species from their nonpathogenic ancestor. The authors use comparative genomic analyses and sequence and structural analyses to accomplish this. This manuscript would be valuable to researchers studying a wide range of pathogenic bacteria within a larger bacterial lineage that contains both pathogens and nonpathogens whose goal is to determine what factors contribute to pathogenicity. Specifically, this manuscript is most useful for nonculturable organisms where functional studies are not possible.

The manuscript is well-written with detailed figures and supplementary data. I appreciated the level

of detail in the methods section and justification of why certain approaches were taken. Suggested edits are provided below, in the order in which they occur in the manuscript.

- Lines 88-92: "Therefore," should be removed, as not all endophytic bacteria that are transmitted by insects are not culturable. Recommend adding the "Therefore," sentence as a third obstacle, as this point differs from the other two. Also recommend adding something describing why point one is an obstacle, such as that it is difficult to study host-pathogen interactions without being able to easily inoculate plants with the pathogens.
- Line 130: What is "DNA polymerases A"? This name was also used throughout the manuscript. DNA polymerase A is eukaryotic. DNA polymerases in prokaryotes have different names.
- Line 180: Recommend adding notes in parentheses that state the abbreviated words for "LC", "SC", and "UT".
- Line 250: Change "nucleotide acids" to "nucleic acids".
- Line 512: Change "relayed to" to "relied".
- Line 513: Change present tense to past tense.
- Line 545: Change "conduced" to "conducted".
- Line 573: Change "stains" to "strains".
- Lines 572-574: Recommend noting somewhere in figure or manuscript which strains are denoted by a star symbol. Currently, only the species is listed, with some strains within that species also being included in the tree but others are absent.
- Line 606: This sentence should be re-written, as it does not make sense as-is.
- Figure 4: Recommend adding legend rather than referring to figure 2 for convenience of the reader.

Staff Comments:

Preparing Revision Guidelines

For complete guidelines on revision requirements, please see the Instructions to Authors at [link to page]. **Submissions of a paper that does not conform to Microbiology Spectrum guidelines will delay acceptance of your manuscript.**

Please return the manuscript within 60 days; if you cannot complete the modification within this time period, please contact me. If you do not wish to modify the manuscript and prefer to submit it to another journal, please notify me of your decision immediately so that the manuscript may be formally withdrawn from consideration by Microbiology Spectrum.

If you would like to submit an image for consideration as the Featured Image for an issue, please contact Spectrum staff.

The authors present novel insights into pathogenicity gene markers of *C. Liberibacter* genus. Those genes will be a main source in pathogenicity studies related to this genus. They go far beyond previous studies in the field. They also found several gene operons lost at the metabolic and other molecular levels. The phylogenetic and genomic analyses show interesting insights into the evolutionary pathway of *C. Liberibacter* and the main differences between pathogenic and non-pathogenic bacteria. Finally, the authors test and validated their hypothesis about host-live style of *C. Liberibacter* species.

Minor commentaries below:

L57. I am not sure about this affirmation. The work in effectors and genome analyses performed by Coaker et al., 2019 “Genome-wide analyses of *Liberibacter* species provides insights into evolution, phylogenetic relationships, and virulence factors” point out to several effectors implied in pathogenicity. Surely, the previous study is not so broad and deep as the present study present here but made an important contribution in the past.

L94-104. Maybe I am wrong... Again, I recommend to review the same research of Coaker and figure out if all the affirmations are still true, for example, L98. “these proteins are mainly present in one strain of the *C.L. asiaticus* species, and not found in other *C.L. asiaticus* strains ...”. As I understand, Coaker et al. (2019) used several strains of CLAs for the effector's prediction and comparisons.

L129. Reference 31. So, from that research, only the sequence of papaya was used?

L407. Include ref 31 and verify if was also in small proteins.

L565. Please attach curated domain profiles.

Point-to-point responses to the comments:

Reviewer comments:

Reviewer #1 (Comments for the Author):

The authors present novel insights into pathogenicity gene markers of *C. Liberibacter* genus. Those genes will be a main source in pathogenicity studies related to this genus. They go far beyond previous studies in the field. They also found several gene operons lost at the metabolic and other molecular levels. The phylogenetic and genomic analyses show interesting insights into the evolutionary pathway of *C. Liberibacter* and the main differences between pathogenic and non-pathogenic bacteria. Finally, the authors test and validated their hypothesis about host-live style of *C. Liberibacter* species.

Minor commentaries below:

L57. I am not sure about this affirmation. The work in effectors and genome analyses performed by Coaker et al., 2019 "Genome-wide analyses of *Liberibacter* species provides insights into evolution, phylogenetic relationships, and virulence factors" point out to several effectors implied in pathogenicity. Surely, the previous study is not so broad and deep as the present study present here but made an important contribution in the past.

Response: There are several fundamental differences between the study conducted by Coaker et al. and our study. First, they aimed to identify the newly transferred genes by comparing Liberibacter species and other members of Rhizobiaceae (non-Liberibacter). They also reported the genes that are uniquely present in each Liberibacter species. However, our genome comparison is designed to identify unique genes that differ between non-pathogenic Liberibacter ancestor and pathogenic decedents. Second, in our study, novel toxin/effector proteins were directly identified through analyzing the genes generated from the method of comparative genomics described above. We determined the novel toxins/effectors based on the following criteria of phylogeny and sequence/structural features: 1) only new genes that were acquired at the ancestor of pathogens; 2) only genes homologous to known toxins/effectors; 3) only genes which display an association with other known toxin domains. By contrast, the toxin/effector identification in the Coaker et al. study is independent of comparative genomics. They rely on a single assumption that toxin/effector proteins should contain a signal peptide or transmembrane region. There are hundreds of proteins containing signal peptides. Furthermore, technical details of how they narrowed down to the 27 candidates were not described in their paper.

Finally, their study did not evaluate false-positives and false-negatives. Therefore, our study and theirs are very different. Most importantly, the three new toxins that we identified were not identified by the Coaker et al study or other earlier studies.

L94-104. Maybe I am wrong... Again, I recommend to review the same research of Coaker and figure out if all the affirmations are still true, for example, L98. "these proteins are mainly present in one strain of the C.L. asiaticus species, and not found in other C.L. asiaticus strains ...". As I understand, Coaker et al. (2019) used several strains of CLas for the effector's prediction and comparisons.

Response: As discussed above, the Coaker et al. study didn't incorporate the concept of comparative genomics in predicting the novel toxins/effectors. They selected the proteins that contain a N-terminal signal peptide or transmembrane region. By analyzing these toxin candidates identified in the Coaker et al. study, we found that they have different phyletic patterns: 14 of them are only present in CLasi species; five of them are present in all Liberibacter species, including the ancestral species, Liberibacter crescens; one is found in CLasi, CLafr, and CLsol; one is found in CLasi and CLeur; one belongs to the prophage components; and eight of them have been removed by recent genome annotations (NC_012985.3). Further, they did not provide any sequence/structure or functional evidence to support that these genes are toxins or effectors. Therefore, we did not discuss those toxin candidates.

Additionally, there are several papers that claimed to have identified the toxins associated with HLB pathogenesis. We have also conducted sequence/structure/phyletic analysis and added the details in the text in response to other reviewers' comments: "Thus far, the potential toxins/effectors identified by earlier studies limit to a small number of HLB-associated pathogens. For instance, CLIBASIA_03875 (m3875) (20) is only present in one C.L. asiaticus strain; other reported toxins, such as Sec-delivered effector 1 (SDE1; CLIBASIA_05315) (21, 22), Las5315mp (23), SDE15 (CLIBASIA_04025) (24), and CLIBASIA_04405 (m4405) (25), are only present in C.L. asiaticus stains, but not in other HLB-associated pathogens (C.L. americanus and C.L. africanus). This suggests that other types of unidentified toxins or effectors might be responsible for the primary HLB pathology."

L129. Reference 31. So, from that research, only the sequence of papaya was used?

Response: Reference 31 was the first study to present the phylogenetic relationship of Liberibacter crescens strain BT-1 and other relatives in the class Alphaproteobacteria. The work

also provided the detailed difference of these bacteria in lifestyle, cell shape, genome size and other characteristics.

L407. Include ref 31 and verify if was also in small proteins.

Response: It is likely that the reviewer indicated a paper other than ref 31 as the ref 31 didn't report any toxin/effector proteins. In our revision, we did include all relevant literatures that have reported the HLB-associated toxins. Please refer to the above response on the earlier identified toxin candidates.

L565. Please attach curated domain profiles.

Response: we have provided the HMM profiles curated in this work as a supplementary Data 2.

Reviewer #2 (Comments for the Author):

This manuscript falls in the category of "re-analyses of large datasets that provide additional insights". The authors present a comprehensive comparative genomic analysis between *Candidatus Liberibacter* genomes both pathogenic and non-pathogenic. Although much of the information in this manuscript might not be novel, the authors present several novel findings that are significant for understanding how *Liberibacter* gains its pathogenicity features and thus based on this the manuscript has its merits. Also, the authors present an overview of the major distinctive features between the genomes of pathogenic and non-pathogenic species, and specifically characterize toxins and prophages, of which both are distinctive for pathogenic species. The section discussing toxins and comparing these toxin classes to other species is very interesting. The manuscript presents a wealth of data, much of it is in the form of supplementary files. The authors overlook some of the important genome features and this needs to be addressed (see below).

As, several comparative genomics studies have already been published, it would have been interesting whether data in this study is congruent with previous findings. For instance, whether the gene lost in the pathogenic *Liberibacter* genomes are similar to results from previous studies (such as in references 59 and 60).

*Response: Although multiple comparative genomics analyses of *Liberibacter* species have been published, our study is distinguished from them by providing several previously undiscovered*

findings including phage diversity, gene loss events, gene gain events, and novel toxins. Among them, the discoveries on gene gains and novel toxins were not reported before.

Regarding the phage diversity, several studies have focused on the identification of some of these phages (and we have cited these papers), but they did not provide a systematic classification and phylogenetic origin analysis.

Regarding the gene loss events, most studies focused on pair-wise comparisons to identify the genes that differ between species, such as the study conducted by Fagen et al. (Ref. 60 in the original version; Ref. 70 in revision), which identified 523 genes that were lost in C.L. solanacearum and C.L. asiaticus when compared to L. crescens. In our study, we analyzed all available genomes and identified 323 orthogroups (335 genes) that were lost in all the pathogens but present in the ancestral Liberibacter crescens. The difference might be due to the technical issues, as Ref. 60 study: 1) only used three genomes; 2) did not conduct an ortholog clustering analysis; 3) did not check the false-positives/false-negatives. That study also conducted KEGG analysis which revealed similar results. However, our operon analysis on the biosynthetic pathways was missing in their work. We have clarified this in the text: "Functional and pathway analysis revealed that biosynthetic pathways of several essential amino acids in the pathogens were disrupted by gene loss, consistent with the earlier studies".

In Ref. 59, Lai et al. (Ref. 69 in revision) utilized a Tn5 transposon mutagenesis screening experiment to identify the genes required for the culture of Liberibacter crescens. It was not designed to identify the gene loss events as many culture-essential genes are not lost in Liberibacter pathogens. However, we do see that 56 of 335 genes that we identified were also found in their list as the culture-essential genes for L. crescens (Ref. 59; Ref. 69 in revision). We have clarified this in the text.

Also, some of the results section are unnecessarily long and very descriptive, it would be better to be succinct in presenting the findings and thus more information can be presented. For instance, the first five sections in the results are very descriptive and get be significantly shortened.

Response: We have tried our best to shorten the main result sections. We sincerely hope that the reviewer could allow us to keep this revised version. This is a top-down inference research work, and we feel it is justified to use the logical inference to dissect the pieces of information before preceding to next step. We hope in this way our explanation will help readers to follow our path of inference when reading the text.

The authors claim recombination and inversions are prominent, but do not explain how and based on what. A lot of the claims are not backed up with the data here.

Response: We have revised and corrected the relevant statement according to the reviewer's comment. The whole genome comparisons show that, in addition to the preserved genome organization, several regions underwent genome inversions, as revealed by the blue lines when the gene arrangement between two genomes is reversed.

The authors discuss recombination as one of the forces driving evolutionary changes whereas as a matter of fact, it is horizontal gene transfer, due to integration of these diverse prophages. The authors do not mention HGT which is a major evolutionary force in fact.

*Response: We agree with the reviewer on this, and we have revised the result section on whole genome comparisons: "We found that between five pathogenic species, the majority of their genome regions preserve similar gene composition and organization, with several large-scale genome inversions (as shown in crossed blue lines). When we compared genomes of the non-pathogenic *L. crescens* and pathogenic *C.L. europaeus*, their genome organization is more diverse than that between genomes of pathogens." Regarding the HGT, we agree with the reviewer and that is also the rationale for us to identify such gene-gain events (HGT) between non-pathogenic ancestor and pathogenic descendants.*

Furthermore, authors do not present any data or mention proteins with signal peptides in the genes that have been acquired in the pathogenic species, and do not discuss any of the secretions systems as potentially part of the acquired genes/clusters driving evolutionary changes in the pathogenic species.

Response: The proteins with signal peptides have been the focus of most recent HLB genomic research, with the aim to discover the potential toxins/effectors. We also conducted a systematic analysis of signal peptide-containing proteins. Unfortunately, both other studies and our own analysis did not lead to identification of any toxins/effectors. This prompted us to realize that focusing on signal peptides is not a good strategy, which is why we came to this comparative phylogeneomic analysis, which has led to the successful identification of novel toxins.

*Regarding the secretion of the potential toxins, we have added a new discussion section to discuss the potential mode of secretion of the toxins in *Candidatus Liberibacter*. "To be noted, only the EEP toxin and several other proteins that were gained during evolution harbor the signal peptide, the export signal of T2SS (48), or the transmembrane (TM) region, which directs*

the protein localization at the bacterial cell membrane so that their coupled toxin module can be exported. However, two other toxin groups, Ntox52 and YdjM, do not contain signal peptide, TM, or other secretion-related domains (48). It is possible that Liberibacter might utilize a yet unknown secretion pathway to export these toxins. Alternatively, these toxins might be exported when these Liberibacter pathogens undergo phage lytic cycle. It has been reported that prophages of the SC type, the one acquired by the ancestor of the pathogens, are activated and convert from lysogenic cycle to lytic cycle, when pathogenic Liberibacter bacteria infect the host plant (41). The lytic cycle of prophages will result in the destruction of bacterial cell membrane, which could release their cytoplasmic molecules including toxins or effectors. Disruption of bacterial cells to release toxins or effectors has been used by bacteria in the situation of kin-selection (72). In the so-called bacteriocin system, the bacteria produce cytoplasmic toxin protein, immunity protein and lysis protein, among which the lysis protein would lyse bacteria to facilitate the release of both the toxin and immunity proteins to promote self-nonself recognition in the bacterial community (72).”

Furthermore, the UT and ZC1 prophages, are two prophages identified here but not well discussed. Is there anything significant about these prophages?

Response: Our classification and phylogenetic analysis suggests that the UT prophages might originate from a duplication of the SC prophage at the common ancestor of all C.L. asiaticus strains, while the ZC1 prophage was likely to have remained following prophage excision from the host genome. Unfortunately, we do not have any clue for their functional significance, but we hope further experiments will be designed to distinguish the roles of these phages.

Please check the English for grammar mistakes, avoid using "it" as an anticipatory pronoun, it can be confusing.

Response: We have carefully revised the manuscript to avoid using “it”.

L81: what is in a similar vein? Please fix. Some of the introduction,

Response: We have changed it to “Likewise” given the context.

L97-L102 is somewhat vague and needs some further work.

Response: we have added the details in the text: “Thus far, the potential toxins/effectors identified by earlier studies limit to a small number of HLB-associated pathogens. For instance, CLIBASIA_03875 (m3875) (20) is only present in one C.L. asiaticus strain; other reported toxins, such as Sec-delivered effector 1 (SDE1; CLIBASIA_05315) (21, 22), Las5315mp (23), SDE15

(CLIBASIA_04025) (24), and CLIBASIA_04405 (m4405) (25), are only present in C.L. asiaticus stains, but not in other HLB-associated pathogens (C.L. americanus and C.L. africanus). This suggests that other types of unidentified toxins or effectors might be responsible for the primary HLB pathology.”

Note, please double check the references. For instance reference 61 is erroneously cited in Line 416-417. I have not checked all the references, this is something the authors will have to do.

Response: Thanks, and we have corrected this citation. We also carefully checked other reference to make sure they are cited correctly.

Some specific inquiries:

L191: fragments of what? Please re-write.

Response: revised: “There is also a small prophage locus (ZC1) on the C.L. solanacearum strain ZC1 genome which might be the ancient prophage remanent after excision from the host genome.”

L208-210: can you elaborate here on this topic, it is not evident.

Response: In this part, we discuss the divergence between the SC prophages in pathogenic Liberibacter. As shown in this study, both LC2 and SC prophages were present in the ancestor of Liberibacter pathogens. Along with the speciation of bacteria, these two prophages should follow a typical vertical evolution; that is, the relationship of the phage descendants should be the same as the relationship of pathogens. Indeed, phylogeny of LC2 terminases mirrors the one of pathogens (Fig. 4A). However, when we examine the SC phage terminases, their relationship didn't follow the tree relationship of pathogens (Fig. 4A). This suggests that the SC phage might have undergone a further diversification and recombination. This is further supported by the whole genome comparisons: During a typical evolution, the prophage descendants should share similarity across the phage region given a similar mutation rate. However, for SC prophages, we observed that similarity of some regions was not detected (Fig. 4B), even though our gene-neighborhood analysis revealed that they are related prophage components. This suggests that a striking divergence is behind these prophages. Our recent research has indeed dissected the evolutionary mechanisms that are behind the prophage diversification, and we are currently preparing a manuscript for publication. In this paper, we want to present the evidence of the unique phage diversification for the sake of a complete genomic analysis. Accordingly, we have revised the parts more precisely: “However, the SC

prophages appear to have undergone further diversification and recombination, given the following facts: 1) their terminases do not follow a typical pattern of vertical evolution, unlike the LC2 terminases (Fig. 4A); 2) in the genome comparisons, the SC phages from different C.L. asiaticus strains display a striking divergence in certain regions, in contrast to other genomic regions (Fig. 4B)."

Fig 4C is not cited in the manuscript.

Response: Thanks, and now we have added citation of Fig. 4C in the result section.

Reviewer #3 (Comments for the Author):

The manuscript entitled "Comparative phylogenomic analysis reveals evolutionary genomic changes and novel toxin families in endophytic *Liberibacter* pathogens" by Tan et al. identifies prophage loci and toxin-associated genes within *Liberibacter* genomes as a means to understand the emergence of endophytic, pathogenic *Liberibacter* species from their nonpathogenic ancestor. The authors use comparative genomic analyses and sequence and structural analyses to accomplish this. This manuscript would be valuable to researchers studying a wide range of pathogenic bacteria within a larger bacterial lineage that contains both pathogens and nonpathogens whose goal is to determine what factors contribute to pathogenicity. Specifically, this manuscript is most useful for nonculturable organisms where functional studies are not possible.

The manuscript is well-written with detailed figures and supplementary data. I appreciated the level of detail in the methods section and justification of why certain approaches were taken. Suggested edits are provided below, in the order in which they occur in the manuscript.

- Lines 88-92: "Therefore," should be removed, as not all endophytic bacteria that are transmitted by insects are not culturable. Recommend adding the "Therefore," sentence as a third obstacle, as this point differs from the other two. Also recommend adding something describing why point one is an obstacle, such as that it is difficult to study host-pathogen interactions without being able to easily inoculate plants with the pathogens.

*Response: Thanks, and we have revised according to the comments: "However, the progress has been slow due to several main obstacles. *Liberibacter* pathogens are endophytic bacteria transmitted naturally by several psyllid vectors such as Asian citrus psyllid (*Diaphorina citri*) (11), African citrus psyllid (*Trioza erytreae*) (12), potato psyllid (*Bactericera cockerelli*) (9), and carrot*

psyllids (Trioza apicalis) (13, 14). This, along with them being unculturable, makes controlled inoculation for studying host-pathogen interaction extremely difficult (15)."

- Line 130: What is "DNA polymerases A"? This name was also used throughout the manuscript. DNA polymerase A is eukaryotic. DNA polymerases in prokaryotes have different names.

Response: Thanks, and we have corrected the error and changed the "DNA polymerase A" to "DNA polymerase I".

- Line 180: Recommend adding notes in parentheses that state the abbreviated words for "LC", "SC", and "UT".

Response: Thanks, and we now added brief descriptions on prophage abbreviations.

- Line 250: Change "nucleotide acids" to "nucleic acids".

Response: corrected.

- Line 512: Change "relayed to "relied".

Response: corrected.

- Line 513: Change present tense to past tense.

Response: corrected.

- Line 545: Change "conduced" to "conducted".

Response: corrected.

- Line 573: Change "stains" to "strains".

Response: We did not locate the "stains" in line 573, but we realize that the reviewer refers the line 673. We have corrected it.

- Lines 572-574: Recommend noting somewhere in figure or manuscript which strains are denoted by a star symbol. Currently, only the species is listed, with some strains within that species also being included in the tree but others are absent.

Response: In lines 572-574, we could not locate the relevant text, but we realize that the reviewer refers the lines 672-674. We provided details of strains in the legend: "Some sequences from different C.L. asiaticus strains are identical and share the same NCBI accession number, so we use one sequence to represent them, indicated by an asterisk symbol

(*). Specifically, *C.L. asiaticus** in B indicates sequences from *C.L. asiaticus* strain psy62, strain gxpsy, strain A4, strain AHCA1, strain Ishi-1, strain JXGC; *C.L. asiaticus** in C indicates sequences from strain A4, strain AHCA1, strain Ishi-1, strain JXGC; and *C.L. asiaticus** in D indicates strain psy62, strain gxpsy, strain A4, strain AHCA1, strain Ishi-1, strain JXGC.”

- Line 606: This sentence should be re-written, as it does not make sense as-is.

Response: The sentence has been revised.

- Figure 4: Recommend adding legend rather than referring to figure 2 for convenience of the reader.

Response: The legend has been added to Fig. 4C.

August 10, 2021

Dr. Dapeng Zhang
Saint Louis University
Biology
St. Louis, MO 63103

Re: Spectrum00509-21R1 (Comparative phylogenomic analysis reveals evolutionary genomic changes and novel toxin families in endophytic *Liberibacter* pathogens)

Dear Dr. Dapeng Zhang:

Thank you for your attention to the reviewers suggestions in your revised manuscript. Your manuscript has been accepted, and I am forwarding it to the ASM Journals Department for publication. You will be notified when your proofs are ready to be viewed.

Sincerely,

Lindsey Burbank
Editor, Microbiology Spectrum

Journals Department
Supplemental Data2: Accept
Supplemental TableS5: Accept
Supplemental Material: Accept

Supplemental Data1: Accept
Supplemental TableS3: Accept